# LEARNING META-FEATURES FOR AUTOML

**Herilalaina Rakotoarison**[1*] **Louisot Milijaona**[2*] **Andry Rasoanaivo**[2]

**Michèle Sebag**[1] **Marc Schoenauer**[1]

[1] TAU, LISN-CNRS–INRIA, Université Paris-Saclay, Orsay, France
[2] MISA, LMI, Université d'Antananarivo, Ankatso, Madagascar

## ABSTRACT

This paper tackles the AutoML problem, aimed to automatically select an ML algorithm and its hyper-parameter configuration most appropriate to the dataset at hand. The proposed approach, MetaBu, learns new meta-features via an Optimal Transport procedure, aligning the manually designed meta-features with the space of distributions on the hyper-parameter configurations. MetaBu meta-features, learned once and for all, induce a topology on the set of datasets that is exploited to define a distribution of promising hyper-parameter configurations amenable to AutoML. Experiments on the OpenML CC-18 benchmark demonstrate that using MetaBu meta-features boosts the performance of state of the art AutoML systems, AutoSkLearn (Feurer et al. 2015) and Probabilistic Matrix Factorization (Fusi et al. 2018). Furthermore, the inspection of MetaBu meta-features gives some hints into when an ML algorithm does well. Finally, the topology based on MetaBu meta-features enables to estimate the intrinsic dimensionality of the OpenML benchmark w.r.t. a given ML algorithm or pipeline. The source code is available at https://github.com/luxusg1/metabu.

## 1 INTRODUCTION

Getting the peak performance of an algorithm portfolio on a particular problem instance is acknowledged a main bottleneck in domains ranging from Constraint Programming and Satisfiability to Machine Learning (Rice, 1976; Hutter et al., 2009; Stern et al., 2010; Kotthoff, 2014; Bergstra et al., 2011; Feurer et al., 2015; Hazan et al., 2018; Fusi et al., 2018; Yang et al., 2019). Early approaches have been investigating the use of general performance models (Rice, 1976), estimating *a priori* the performance of any algorithm on any problem instance, where each problem instance is described by a vector of so-called *meta-features*, and the performance model is learned in this meta-feature space.

In the context of supervised Machine Learning, many meta-features have been manually designed to describe datasets (Caliński & Harabasz, 1974; Vilalta, 1999; Bensusan & Giraud-Carrier, 2000; Pfahringer et al., 2000; Peng et al., 2002; Ali & Smith, 2006; Song et al., 2012; Bardenet et al., 2013; Feurer et al., 2014; 2015; Pimentel & de Carvalho, 2019; Lorena et al., 2019). After a series of international AutoML challenges, aimed to automating the selection and tuning of ML pipelines[1] (Hutter et al., 2019; Guyon et al., 2019), it seems that a general accurate performance model can hardly be based on these meta-features (Misir & Sebag, 2017) (Section 2): for instance the challenge-winner AutoSkLearn (Feurer et al., 2015) relies on Bayesian optimization and iteratively learns and exploits one performance model specific to the dataset at hand; PMF (Fusi et al., 2018) uses a probabilistic collaborative filtering approach, where the cold-start problem is handled as in AutoSkLearn; OBOE (Yang et al., 2019) likewise uses a collaborative filtering approach, combined with active learning.

Nevertheless, the definition of good meta-features remains desirable for two reasons. The first motivation remains to achieve AutoML with a decent performance *vs* cost trade-off. Relevant

---

[*] Equal contribution (herilalaina.rakotoarison@inria.fr and milijaonalouisot@gmail.com).
[1] An ML pipeline consists of a data preparation stage followed by the model learning stage. Each stage involves a number of options and a varying number of hyper-parameters, depending on the former selected options. Terms *ML pipeline* and *ML algorithm* will be used interchangeably in the remainder of the paper.

meta-features are expected to define a reliable *topology* on the dataset space, such that two datasets are close iff the best hyper-parameter configurations for these datasets are close. Such a topology would support an inexpensive and efficient AutoML strategy: selecting the best hyper-parameter configurations of the nearest neighbor(s) of the current dataset.

The second motivation is to better understand the dataset space w.r.t. a given ML algorithm, to estimate its intrinsic dimension and to appreciate the distribution of the ML benchmark suites thereon.

This paper presents the *Meta-learning for Tabular Data* (METABU) approach, formalizing and tackling the construction of good meta-features relatively to an ML algorithm $\mathcal{A}$ as an Optimal Transport (OT) problem (Cuturi, 2013; Peyré & Cuturi, 2019). Formally, METABU considers two representations of the datasets: the basic one consists of 135 manually designed meta-features (Appendix E). The target one, out-of-reach except  for the datasets in the benchmark suite, represents a dataset as the distribution of the hyper-parameter configurations of $\mathcal{A}$ yielding the top performances for this dataset. Optimal Transport is used to find a linear transformation of the basic meta-features, such that the resulting Euclidean distance emulates the Wasserstein-Gromov distance (Mémoli, 2011) on the target representation (Section 3). Overall, Metabu learns once for all new meta-features, aimed to capture the topology and neighborhoods corresponding to the target representation. These meta-features can be computed from scratch for each new dataset. This approach contrasts with Yang et al. (2019) and Fusi et al. (2018) that both require a cold-start phase, launching configurations for each new dataset and using their performance to find the representation of the new dataset.

The contribution of METABU is threefold. Firstly, the METABU meta-features define an efficient topology, that can be used to sample the most promising hyper-parameter region for new datasets. Secondly, the relevance of these meta-features is demonstrated as they can be used as representation space to initialize AutoSkLearn (Feurer et al., 2014) and PMF (Fusi et al., 2018): the hybrid approaches AutoSkLearn+METABU and PMF+METABU, are shown to significantly outperform AutoSkLearn and PMF on the OpenML CC-18 (Bischl et al., 2019) benchmark. Lastly, the approach provides some hints into the AutoML problem, enabling to estimate the intrinsic dimensionality (Facco et al., 2017) of the dataset space w.r.t. an ML algorithm: the higher the dimensionality, the more complex the algorithm. It is interesting to compare the intrinsic dimensions of the OpenML CC-18 w.r.t. AutoSkLearn (Feurer et al., 2015), SVM (Boser et al., 1992), or Random Forest (Breiman, 2001). Furthermore, the METABU meta-features can be inspected and confirm some "tricks of the trade" about when an algorithm does well.

The paper is organized as follows. Section 2 briefly discusses related work and introduces OT formal background for the sake of self-containedness. Section 3 describes the METABU algorithm. In Section 4, the merits of the METABU meta-features are empirically demonstrated on configuration selection and optimization tasks, comparatively to the state of the art. Lastly, we discuss how the METABU meta-features provide an interpretable description of the *niche* of the considered ML algorithms.

## 2 RELATED WORK AND FORMAL BACKGROUND

**AutoML & meta-features**   Most ML meta-features (Caliński & Harabasz, 1974; Vilalta, 1999; Bensusan & Giraud-Carrier, 2000; Pfahringer et al., 2000; Peng et al., 2002; Ali & Smith, 2006; Song et al., 2012; Bardenet et al., 2013; Feurer et al., 2015; 2014; Pimentel & de Carvalho, 2019; Lorena et al., 2019) have been manually designed to describe supervised datasets based on descriptive statistics, information theory (quantifying relationships among features/labels), geometrical structure of the dataset, and landmarking (performance of cheap classifiers such as linear discriminant and decision trees). In the neighbor fields of Satisfiability or Constraint Programming, circa one hundred meta-features have also been manually designed (Nudelman et al., 2004; Xu et al., 2008). In contrast to the efficiency of SAT or CP meta-features however (Kotthoff, 2014), the AutoML search can hardly rely on the only metric defined from the ML meta-features after (Misir & Sebag, 2017; Muñoz et al., 2018); in practice, they are often used to initialize the optimization search (Feurer et al., 2015).

Another approach is to *learn* meta-features, e.g. by making strong assumptions on the performance model (Hazan et al., 2018) or by leveraging distributional neural networks (de Bie et al., 2019; Maron et al., 2020). In the latter case, these meta-features are functions of the dataset distribution and consist of the last layer of a distributional NN trained in view of a particular task. Dataset2Vec (Jomaa et al., 2021) learns meta-features to detect whether two data patches (subset of samples described by a subset of features) are extracted from the same whole dataset. OTDD (Alvarez-Melis & Fusi,

2020) uses OT to learn a mapping over the joint feature and label spaces. A significant drawback of distributional neural network approaches, limiting their ability to handle general tabular datasets (with widely varying number of features, missing values, heterogeneous variables) is due to the shortage of training (meta)-samples. Neural networks notoriously need large amounts of samples to be efficiently trained, while AutoML benchmarks include less than a hundred datasets. For this reason, the proposed METABU approach proceeds by building upon existing meta-features.

**Optimal Transport** Let $(\Omega_x, d_x)$ and $(\Omega_y, d_y)$ denote compact metric spaces, and **x** and **y** distributions[2] respectively defined on $\Omega_x$ and $\Omega_y$. The search space $\Gamma(\mathbf{x}, \mathbf{y})$ is the space of all distributions on $\Omega_x \times \Omega_y$ with marginals **x** and **y**. Let the transport cost function $c : \Omega_x \times \Omega_y \mapsto \mathbb{R}^+$ be a scalar function on $\Omega_x \times \Omega_y$[3].

The OT problem consists in finding a distribution in $\Gamma(\mathbf{x}, \mathbf{y})$ yielding a minimal transport cost expectation (Peyré & Cuturi, 2019); this minimal transport cost expectation defines the Wasserstein distance of **x** and **y**: $d_W^q(\mathbf{x}, \mathbf{y}) = \min_{\gamma \in \Gamma(\mathbf{x}, \mathbf{y})} \mathbb{E}_{(x,y) \sim \gamma}[c^q(x,y)]^{1/q}$, with $q$ a positive real number, set to 1 in the following.

Another OT-based distance is the Gromov-Wasserstein distance (GW) (Mémoli, 2011), measuring how well a distribution in $\Gamma(\mathbf{x}, \mathbf{y})$ preserves the distances on both $\Omega_x$ and $\Omega_y$, akin a rigid transport between both domains: $d_{GW}^q(\mathbf{x}, \mathbf{y}) = \min_{\gamma \in \Gamma(\mathbf{x}, \mathbf{y})} \mathbb{E}_{(x,y) \sim \gamma, (x'y') \sim \gamma}[|d_x(x, x') - d_y(y, y')|^q]^{1/q}$.

The Fused Gromov-Wasserstein (FGW) distance (Titouan et al., 2019) combines both these distances.

**Definition 1** *The Fused q-Gromov-Wasserstein distance is defined on $\Omega_x \times \Omega_y$ as follows:*

$$
\begin{aligned}
d_{FGW;\alpha}^q(\mathbf{x}, \mathbf{y}) \quad = \min_{\gamma \in \Gamma(\mathbf{x}, \mathbf{y})} (1-\alpha) \underbrace{\left( \int_{\Omega_x \times \Omega_y} c^q(x,y) \mathrm{d}\gamma(x,y) \right)^{\frac{1}{q}}}_{\textit{Wasserstein Loss}} \\
+ \alpha \underbrace{\left( \int_{\Omega_x \times \Omega_y} \int_{\Omega_x \times \Omega_y} |d_x(x,x') - d_y(y,y')|^q \mathrm{d}\gamma(x,y)\mathrm{d}\gamma(x',y') \right)^{\frac{1}{q}}}_{\textit{Gromov-Wasserstein Loss}}
\end{aligned}
\tag{1}
$$

$\alpha \in [0,1]$ *is a trade-off parameter: For $\alpha = 0$ (resp. $\alpha = 1$), the fused q-Gromov-Wasserstein distance is exactly the q-Wasserstein distance $d_W^q$ (resp. the q-Gromov-Wasserstein distance $d_{GW}^q$).*

The Wasserstein distance and variants thereof have been successfully used to evaluate the "alignment" among datasets, e.g. between the source and the target datasets in the context of domain adaptation (Courty et al., 2017) or transfer learning (Alvarez-Melis & Fusi, 2020). FGW distance has been used to enforce the consistency of the latent space when jointly training several Variational Auto-Encoders (Xu et al., 2020; Nguyen et al., 2020). METABU will likewise take inspiration from OT to create a bridge between two representations of the datasets: the basic one, and the target one, critically using both GW and FGW distances.

## 3 OVERVIEW OF METABU

Let $\mathcal{A}$ and $\Theta_{\mathcal{A}}$ respectively denote an ML pipeline and its hyper-parameter configuration space; subscript $\mathcal{A}$ is omitted when clear from the context. Space $\Theta$ is embedded into the $a$-dimensional real-valued space $\mathbb{R}^a$, using a one-hot encoding of Boolean and categorical hyper-parameters. After describing the principle of the approach, some key issues are detailed: the augmentation of the AutoML benchmark to avoid overfitting, and the setting of the number $d$ of the METABU meta-features, estimated from the intrinsic dimensionality of the AutoML benchmark suite.

---

[2]Distributions will be denoted in boldface

[3]When $\Omega_x = \Omega_y = \Omega$, unless otherwise stated, the transport cost $c(x,y)$ is the Euclidean distance $d(x,y)$.

**Principle.** Intuitively, two representations can be associated with a dataset: The *basic representation* $x \in \mathbb{R}^D$ of a dataset reports the values of the $D$ manually designed meta-features for this dataset. By construction, it can be cheaply computed for any dataset. The *target representation* **z** of a dataset is the distribution on the space $\Theta$ supported by the configurations yielding the best performances on this dataset. This precious target representation is unreachable in practice, but can be approached after the performances of the models learned with a number of configurations (aka configuration performances) have been assessed. In practice, the configuration performances are only available for a small number $n$ of datasets (more below). The difference between the basic and the target topologies is depicted on Fig. 1, in $\Theta$ space (projected on first two PCA eigenvectors).

In order to build a bridge between both representations, let us consider an intermediate representation derived from the target representation, mapping each $(z_i)_{1 \le i \le n}$ on some $u_i \in \mathbb{R}^d$ using a distance-preserving projection, e.g. Multi-Dimensional Scaling (MDS) (Cox & Cox, 2001). METABU tackles an Optimal Transport problem so as to learn a mapping $\psi : \mathbb{R}^D \mapsto \mathbb{R}^d$ from the basic representation on the projected target representation space such that the $\psi(x_i)_{1 \le i \le n}$ are aligned with the $u_i$s in the sense of the q-Fused Gromov-Wasserstein distance (Section 2). In brief, mapping $\psi$ sends the basic meta-feature space on $\mathbb{R}^d$, such that the Euclidean metric on the $\psi(x_i)$ reflects the Euclidean metric on the $u_i$s, itself reflecting the metric on the target $z_i$s. The descriptive features of the $\psi(x_i)$s,

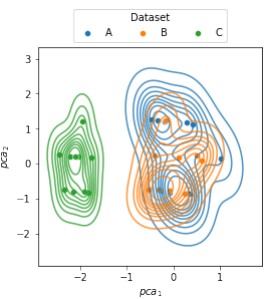

Figure 1: Top configurations of datasets $A$, $B$, and $C$, where $B$, in orange (resp. $C$, in green) is the nearest neighbor of $A$ w.r.t. target (resp. basic) representation.

referred to as METABU meta-features, are meant to both be cheaply computable from the basic meta-features, and define a Euclidean distance conducive to the AutoML task.

**Augmenting the AutoML benchmark.** The OpenML CC-18 (Bischl et al., 2019), to our knowledge the largest curated tabular dataset benchmark (that will be used in the experiments), contains $n = 72$ classification datasets; the target representation is available for 64 of them. The shortage of such datasets yields a risk of overfitting the learned meta-features. This challenge is tackled by augmenting the OpenML CC-18 benchmark suite, using a bootstrap procedure (Efron, 1979).[4] The goal is to pave the meta-feature space more densely and more accurately than through e.g., perturbing the basic representation with Gaussian noise (the visualization of the augmented benchmark is displayed on Fig. 6, Appendix A).

**The algorithm** The algorithm is provided the $p = 1,000 \times n$ training datasets of the benchmark suite, augmented as described above (pseudo-code in Appendix B). The METABU meta-features are constructed in a 3-step procedure, illustrated on Fig. 2:

**Step 1**: *Target representation and Wasserstein distance*. Considering the $i$-th training dataset, let $\Theta_i \subset T$ denote the set of hyper-parameter configurations with performance in the top-$L$ known configuration performances ($L = 20$ in the experiments).[5]
The target representation $\mathbf{z}_i$ of the $i$-th dataset is the discrete distribution with support $\Theta_i$. The distance $d_W^1(\mathbf{z}_i, \mathbf{z}_j)$ is the 1-Wasserstein distance among distributions (Section 2).

**Step 2**: *Projecting the target representation on $\mathbb{R}^d$*. The second step consists in projecting the $\mathbf{z}_i$s on $\mathbb{R}^d$, where $d$ is identified using an intrinsic dimensionality procedure (details below), using Multi-Dimensional Scaling (Cox & Cox, 2001), such that the distance $d(u_i, u_j)$ approximates the 1-Wasserstein distance $d_W^1(\mathbf{z}_i, \mathbf{z}_j)$ (Fig. 2, leftmost and second subplots). Note that by construction, the $u_i$s are defined up to an isometry on $\mathbb{R}^d$.

---

[4]For each $\ell$-size dataset $E$ in the benchmark suite, $K = 1,000$ new datasets $F_1, \ldots F_K$ are generated, where $F_i$ includes $\ell$ examples selected in $E$ uniformly with replacement. The basic representation of $F_i$ is computed, and its target representation is set to that of $E$.

[5]Early attempts to define $\Theta_i$ in a more sophisticated way, e.g. using t-test to distinguish the "good" configurations from the others, led to an uninformative target representation. A tentative interpretation for this fact is that quite a few OpenML datasets are very easy, leading to retain all configurations for these datasets and blurring the target representation.

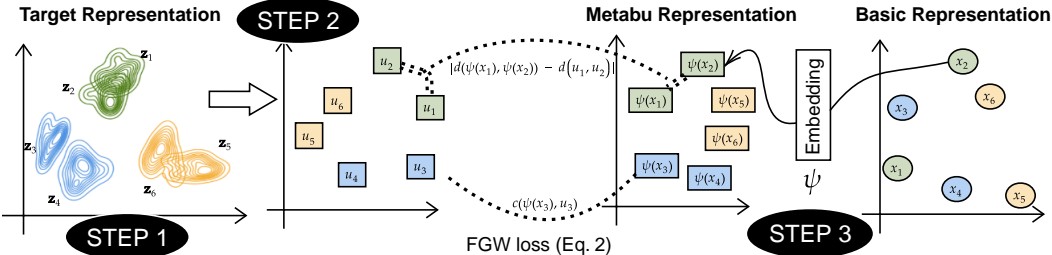

Figure 2: From basic to METABU meta-features using Fused Gromov-Wasserstein. Basic (respectively METABU) representations are depicted by circles (resp. squares). Target representations are depicted in the leftmost subplot. Neighbor datasets in the target space have same color in all subplots.

**Step 3**: *Learning the* METABU *meta-features*. Let $\mathbf{x} = \frac{1}{p}\sum_{i=1}^{p}\delta_{x_i}$ denote the uniform discrete distribution on $\mathbb{R}^D$ whose support is the set of $p$ datasets using their basic representations.
Let $\mathbf{u} = \frac{1}{n}\sum_{i=1}^{n}\delta_{u_i}$ denote the uniform discrete distribution on $\mathbb{R}^d$ whose support is the set of $u_i$s defined above. The METABU meta-feature space is built by finding a mapping $\psi$ from $\mathbb{R}^D$ on $\mathbb{R}^d$ that pushes the representation metric on $\mathbb{R}^d$, that is, such that the image of $\mathbf{x}$ via $\psi$ is as close as possible to $\mathbf{u}$, and reflects its topology in the FGW sense (Fig. 2, rightmost and third subplots).

Formally, let $\psi_{\sharp}\mathbf{x} \overset{\text{def}}{=} \frac{1}{p}\sum_{i=1}^{p}\delta_{\psi(x_i)}$ be the push-forward distribution of $\mathbf{x}$ on $\mathbb{R}^d$ for a given $\psi$. The overall optimization problem is to find a mapping $\psi^*$ that minimizes the FGW distance between the $\mathbf{u}$ distribution and the push distribution $\psi_{\#}^*\mathbf{x}$:

$$\psi^* = \underset{\psi \in \Psi}{\arg\min}\ d_{FGW;\alpha}\left(\psi_{\sharp}\mathbf{x}, \mathbf{u}\right) + \lambda\|\psi\| \tag{2}$$

with $\lambda$ the regularization weight and $\|\psi\|$ the norm of the $\psi$ function. Note that, as $\mathbf{u}$ and $\psi_{\#}\mathbf{x}$ are distributions on the same space $\mathbb{R}^d$, the transport cost $c$ is the Euclidean distance on $\mathbb{R}^d$.

In the following, only linear mappings $\psi$ are considered for the sake of avoiding overfitting and facilitating the interpretation of the METABU meta-features w.r.t. the manually designed meta-features. The norm of $\psi$ is set to the $L_1$ norm of its weight vector.

Taking inspiration from Xu et al. (2020), the efficient optimization of Eq. 2 is achieved using a bilevel optimization formulation. For a given $\psi$, the inner optimization problem consists of minimizing $d_{FGW,\alpha}(\psi_{\sharp}\mathbf{x}, \mathbf{u})$ (Eq. 1). This problem is solved using a proximal gradient method (Xu et al., 2019), along an iterative approach: given an estimation of the transport map $\gamma^{(t)}$, a sub-problem is defined to refine $\gamma$, it is solved using the Sinkhorn algorithm (Cuturi, 2013), and its solution is used to compute $\gamma^{(t+1)}$ (the number of iterations is set to 10 in the experiments).
The outer optimization problem consists of optimizing $\psi$: The transport matrix $\gamma$ is treated as a constant, and the outer objective function (Eq. 2) is solved with ADAM optimizer (Kingma & Ba, 2015) with learning rate 0.01, $\alpha = 0.5$ and $\lambda = 0.001$.

**Intrinsic dimension of the space of datasets** The main hyper-parameter of METABU is the number $d$ of meta-features needed to approximate the target representation. Indeed, $d$ depends on the considered algorithm $\mathcal{A}$: the more diverse the target representations associated with datasets, the harder the AutoML problem, the higher $d$ needs to be. In the other extreme case (all datasets have similar target representations), the AutoML problem becomes trivial.

To our best knowledge, measuring the intrinsic dimension of the dataset space w.r.t. a learning algorithm has not been tackled in the literature. The approach proposed to do so builds on Levina & Bickel (2005) and (Facco et al., 2017), exploiting the fact that the number of points in a hypersphere of radius $r$ in dimension $d$ increases like $r^d$. Formally, to each sample $x$ is associated its first and second nearest neighbors, respectively noted $x^{(1)}$ and $x^{(2)}$ and let $\mu(x) = \frac{d(x,x^{(2)})}{d(x,x^{(1)})}$ be the ratio of their distances to $x$. With no loss of generality, the samples are ordered by increasing value of $\mu$ (i.e., $\mu(x_i) \leq \mu(x_j)$ for all $i < j$). Let $d$ be the slope of the linear approximation of the 2D curve defined by $\{(log(\mu(x_i)), -log(1 - \frac{i}{m+1}), 1 \leq i \leq m\}$. Then $d$ provides a guaranteed approximation of the

intrinsic dimensionality of the manifold where the $x_i$s family lives (Facco et al., 2017).

It is commonplace to say that the good distance between any two items depends on the considered task. The original approach used in METABU in order to estimate the intrinsic dimensionality of the dataset space, is to set the distance of two datasets to the 1-Wasserstein distance among their target representations.

## 4 EXPERIMENTS

All materials (code, data, and instructions) are made available at https://github.com/luxusg1/metabu. Runtimes are measured on an Intel(R) Xeon(R) CPU E5-2660 v2 @ 2.20GHz.

### 4.1 EXPERIMENTAL SETTINGS

**Goals of experiment.** The first goal is to assess the dataset neighborhoods induced by the METABU meta-features (constructed on the top of the manually designed 135 meta-features from the literature) and the relevance of these dataset neighborhoods w.r.t. the AutoML problem. The performances are assessed against three baselines: AutoSkLearn meta-feature set (Feurer et al., 2014), Landmark (Pfahringer et al., 2000) and SCOT (Bardenet et al., 2013) meta-feature sets. All meta-feature sets are detailed in Appendix E. For Tasks 2 and 3 (see below), an additional baseline is based on the uniform sampling of the hyper-parameter configuration space, for sanity check.

The second goal of experiments is to assess the sensitivity of METABU w.r.t. its own two hyper-parameters, the weight $\alpha$ used to balance the importance of the Wasserstein and Gromov-Wasserstein distances in FGW (Eq. 1), and the regularization weight $\lambda$ involved in the optimization of $\psi$ (Eq. 2). The third goal is to gain some understanding of the dataset landscape, and see whether the METABU meta-features give some hints into when a given ML algorithm or pipeline does well (its *niche*).

**Performance indicators.** Three tasks are considered to investigate the relevance of the METABU meta-features. The performance indicators are measured using a Leave-One-Out process (detailed in Appendix C).

**Task 1**: *Capturing the target topology*. For each test dataset, one considers its nearest neighbors w.r.t. the target topology (the 1-Wasserstein metric on the target representation), and its nearest neighbors w.r.t. the Euclidean distance on the METABU and meta-feature sets. The alignment between both ordered lists is measured using the normalized discounted cumulative gain over the first $k$ neighbors (NDCG@k) (Burges et al., 2005), with $5 \leq k \leq 35$. The performance indicator is the NDCG@k averaged on test datasets.

**Task 2**: *AutoML with no performance model (Initialization)*. For each test dataset and each meta-feature set $mf$, let $\hat{\mathbf{z}}_{mf}$ be the distribution on the considered hyper-parameter configuration space:

$$\hat{\mathbf{z}}_{mf} = \frac{1}{Z} \sum_{\ell=1}^{10} exp(-\ell)\, \mathbf{z}_\ell$$

where $\mathbf{z}_\ell$ is the target representation of the $\ell$-th neighbor of the dataset w.r.t. Euclidean distance on the $mf$ space, and $Z$ a normalization constant. This distribution $\hat{\mathbf{z}}_{mf}$ is used to iteratively and independently sample the hyper-parameter configurations, and the performances of the learned models are measured. Letting $r(t, mf)$ denote the rank of the performance associated with meta-feature set $mf$ after $t$ iterations, the performance curves report $r(t, mf)$ for the METABU and baseline meta-feature sets (plus a uniform hyper-parameter configuration sampler for sanity check), averaged over the test datasets.

**Task 3**: *AutoML with performance model (Optimization)*. AutoML systems based on performance models cannot be directly compared with METABU as they acquire additional information along the AutoML search: they iteratively use a performance model to select a hyper-parameter configuration, and update the performance model using the performance of the selected configuration. In Task 3, the relevance of meta-feature sets is investigated in that they govern the initialization for AutoSkLearn and PMF performance models. The performance indicator is the rank of the performance obtained by AutoSkLearn using METABU meta-features to initialize its performance model,

noted METABU+AutoSkLearn (respectively, the rank of the performance of PMF using METABU meta-features to initialize its performance model, noted METABU+PMF).

The difference between Tasks 2 and 3 can be viewed in terms of *Exploration* vs *Exploitation*: getting a good performance on Task 2 requires to identify a sweet configuration spot for each dataset (*Exploitation*). Quite the contrary, getting a good performance on Task 3 requires to identify a sufficiently good and diverse configuration region, such that the search initialized in this region, gathering additional information about the performance of new configurations on the current dataset along time, eventually yields an even better configuration (*Exploration*).

**Benchmarks.** The considered AutoML benchmark is the OpenML Curated Classification suite 2018 (Bischl et al., 2019), including 72 binary or multi-class datasets out of which 64 have enough learning performance data to give a good approximation of their target representation. The performance indicators are measured using Leave-One-Out (details in Appendix C). The basic meta-features are computed for each dataset using the open source library PyMFE (Alcobaça et al., 2020).

METABU is validated in the context of three ML algorithms: Adaboost (Freund & Schapire, 1997), RandomForest (Breiman, 2001) and SVM (Boser et al., 1992), using their scikit-learn implementation (Pedregosa et al., 2011); and two AutoML pipelines, AutoSkLearn (Feurer et al., 2015) and PMF (Fusi et al., 2018). The associated hyper-parameter configuration spaces are detailed in Appendix D.

For Adaboost, RandomForest and SVM, the target representation of each training dataset is based on the top-20 configurations in OpenML (out of 37,289 for Adaboost, 81,336 for RandomForest and 37,075 for SVM), initially generated by van Rijn & Hutter (2018). For AutoSkLearn, the target representation is generated from scratch, running 500 configurations per training dataset and retaining the top-20. For PMF, the top-20 configurations are extracted from the collaborative filtering matrix for each training dataset (Fusi et al., 2018).

### 4.2 COMPARATIVE EMPIRICAL VALIDATION OF METABU

The performances of METABU and the baselines on the three tasks are displayed on Fig. 3. The overall CPU cost on Task 2 (resp. Task 3) is circa 1,900 (resp. 2,300) hours (full runtimes in Fig. 7). Appendix H reports the detailed results in Tables 6,7 and 8, indicating the confidence level of the results after a Wilcoxon rank-sum test for performances and Mann Whitney Wilcoxon test for ranks.

**Task 1**: *Capturing the target topology*, Fig. 3a. The results show that the metric based on the METABU meta-features better matches the target topology than the metric based on the baseline meta-feature sets, all the more so as the number $k$ of nearest neighbors increases. The higher variance of NDCG@k for METABU is explained as the metric depends on the meta-feature training, while the metrics based on the baselines are deterministic. As could be expected, this variance decreases with $k$. Despite this variance, METABU significantly outperforms all baselines for all $k$ and all hyper-parameter configuration spaces.

**Task 2**: *AutoML with no performance model (Initialization)*, Fig. 3b. All rank curves start at 3, as five hyper-parameter configuration samplers are considered. For RandomForest, the sampler based on the SCOT meta-feature set dominates in the first 5 iterations, and remains good at all time; METABU dominates after the beginning; all other approaches but the uniform sampler yield similar performances. For Adaboost, the sampler based on the AutoSkLearn meta-feature set dominates in the first 3 iterations, and METABU is statistically significantly better than all other approaches thereafter. For SVM, METABU very significantly dominates all other approaches.

**Task 3**: *AutoML with performance model (Optimization)*, Fig. 3c. In first time steps (left of the dashed bars), the performance models of AutoSkLearn or PMF are initialized using the performances of the hyper-parameter configurations sampled as in Task 2; in the following time steps, the hyper-parameter configurations are sampled using the performance model. The most striking result is that the METABU+AutoSkLearn rank improves on that of AutoSkLearn (Fig. 3c, left) although they only differ in the initialization of the performance model, and the AutoSkLearn meta-feature set is optimized to Task 3. Likewise, the rank of METABU+PMF improves on that of PMF (Fig. 3c, right). The comparison also involves Random2× and Random4× uniform samplers, respectively returning the best performance out of 2 or 4 uniformly sampled configurations (Fusi et al., 2018); METABU+PMF significantly improves on Random4× after the 10th iteration. This suggests that on

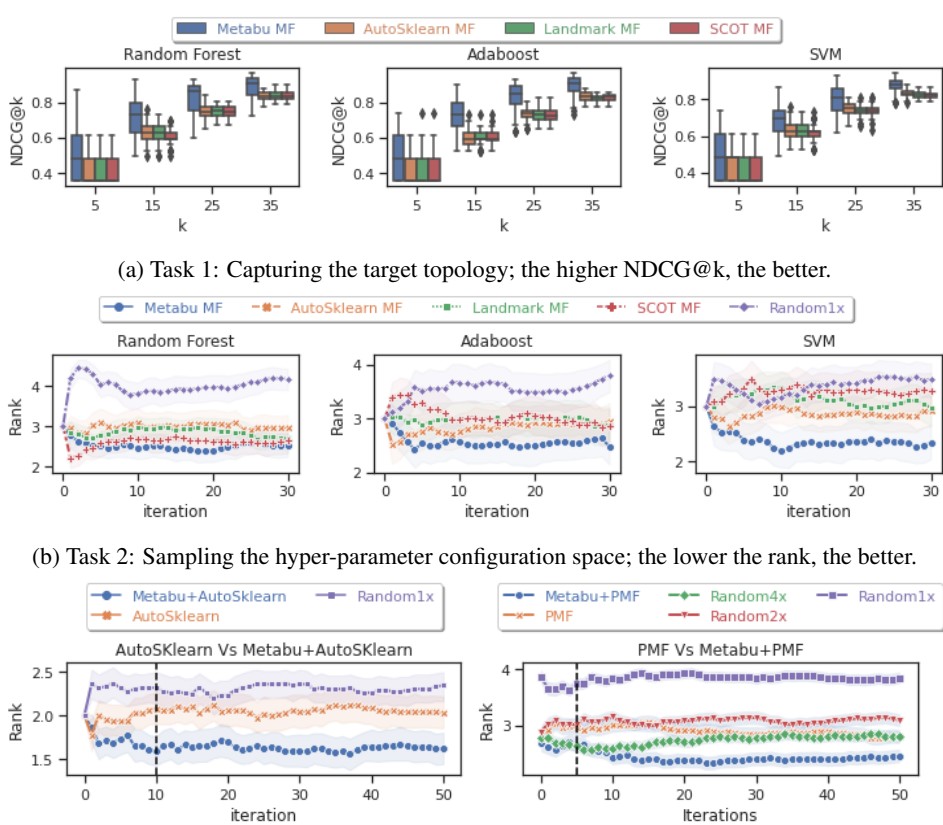

(a) Task 1: Capturing the target topology; the higher NDCG@k, the better.

(b) Task 2: Sampling the hyper-parameter configuration space; the lower the rank, the better.

(c) Task 3: Initializing a performance model to sample the hyper-parameter configuration space.

Figure 3: Empirical assessment of METABU meta-features comparatively to the baselines meta-feature sets and uniform hyper-parameter sampling (better seen in color).

the OpenML benchmark, the METABU meta-features efficiently enable both to passively sample the hyper-parameter configuration space, and to retrieve the configurations best appropriate to update the performance model and explore good regions of the space.

### 4.3 SENSITIVITY ANALYSIS

The own two hyper-parameters of METABU are the $\alpha$ trade-off parameter between Wasserstein and Gromov-Wasserstein distance (Eq. 1) and the regularization weight $\lambda$ (Eq. 2). The sensitivity of METABU w.r.t. both parameters is investigated on Task 1, by inspecting the difference NDCG@10(METABU) - NDCG@10(AutoSkLearn) for $\alpha$ ranging in $\{0.1, 0.3, 0.5, 0.7, 0.99\}$ and $\lambda$ in $\{10^{-1}, \ldots, 10^{-4}\}$. The result, displayed in Fig. 4, shows that the difference is positive in the whole considered domain, with NDCG@10(METABU) statistically significantly better than NDCG@10(AutoSkLearn) according to Student t-test with p-value 0.05.

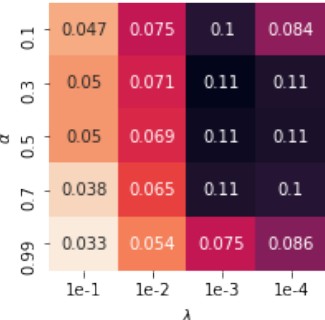

Figure 4: METABU: Sensitivity of NDCG@10 w.r.t. $\alpha$ and $\lambda$, comparatively to AutoSkLearn (darker is better).

Interestingly, a low sensitivity of METABU is observed w.r.t. the regularisation weight $\lambda$, provided that it is small enough ($\lambda \leq 10^{-3}$). For such small $\lambda$ values, a low sensitivity is also observed w.r.t. $\alpha$ in a large range ($.3 \leq \alpha \leq .7$). This result confirms the importance of taking into account both the Wasserstein and Gromov-Wasserstein distances on the

target representation space: discarding the former ($\alpha \leq .1$) or the latter ($\alpha \geq .99$) significantly degrades the performance, and the performance is stable in the $[.3, .7]$ region.

### 4.4 Toward understanding the dataset landscape

A first original result is to provide a principled estimate of the intrinsic dimension of the dataset space w.r.t. the considered ML algorithms. As detailed in Appendix G.1 with a stability analysis, the intrinsic dimension $d$ of the OpenML benchmark is circa 6 for AutoSkLearn, 8 for Adaboost, 9 for RandomForest and 14 for Support Vector Machines. As $d$ reflects by construction how diverse the datasets are w.r.t. the ML algorithm, it is interesting to see that the most flexible AutoSkLearn ML pipeline corresponds to the lowest intrinsic dimension.

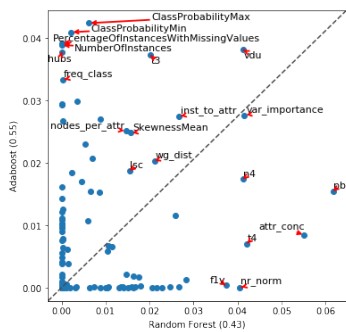

Figure 5: Comparative importance of meta-features for RandomForest (x-axis) and Adaboost (y-axis).

METABU also delivers some insights into what matters in the dataset landscape, and why a given algorithm should behave better than another on a particular dataset, as follows. The images $\psi(x_i)$ of datasets according to the METABU meta-features learned in the context of an algorithm $\mathcal{A}$ are processed using PCA, and the importance of a manually designed meta-feature is measured from the norm of its projection $i_{\mathcal{A}}(mf)$ on the first PCA axis.

Two ML algorithms or pipelines $\mathcal{A}$ and $\mathcal{B}$ can thus be visually compared, by plotting each meta-feature as a 2D point with coordinates $(i_{\mathcal{A}}(mf), i_{\mathcal{B}}(mf))$. As shown on Fig. 5, with respectively $\mathcal{A}$ set to RandomForest and $\mathcal{B}$ to Adaboost, one sees that actually few features matter for both RandomForest and Adaboost (the features nearest to the upper right corner), mostly the Dunn index (Dunn, 1973) and the features importance. Some findings reassuringly confirm the practitioner's expertise: the percentage of instances with missing values matters much more for Adaboost than for RandomForest; the class imbalance (ClassProbabilityMax and ClassProbabilityMin) matters for Adaboost. Complementary results (Appendix G.2) show that the sparsity of the data matters for Support Vector Machines. Some other findings are less expected, e.g. the importance of the data density, minimal skewness and kurtosis for AutoSkLearn; these findings are tentatively explained from the fact that AutoSkLearn includes classifiers such as Linear Discriminant or Logistic Regression.

## 5 Conclusion and Perspectives

METABU provides an algorithm-dependent way to achieve AutoML, through learning meta-features as linear combinations of the manually designed meta-features of the literature, optimized to capture both the top configurations for the datasets and their topology, via preserving their Wasserstein and Gromov-Wasserstein distances. The efficiency of the approach is empirically demonstrated as the METABU meta-features contribute to outperform strong baselines, including AutoSkLearn (Feurer et al., 2014) and PMF (Fusi et al., 2018).

An interesting side-product of the approach is to shed some light on the complexity of the AutoML problem, by estimating the intrinsic dimension of the dataset landscape. Surprisingly, this intrinsic dimension is relatively modest ($< 14$). While this result is comforting when considering the small number of datasets in the AutoML benchmarks, it should however be taken with a grain of salt: the intrinsic dimension might merely reflect the specifics of the OpenML benchmark, as the datasets might have been selected over the years to provide evidence for the merits of mainstream ML algorithms while discarding too hard datasets.

A perspective for further research is to assess the validity of the proposed meta-features and the stability of intrinsic dimensions on other AutoML benchmarks: the underlying question is to which extent AutoML, too, is prone to overfitting.

Another perspective is to exploit METABU to conduct a comprehensive empirical assessment of a new algorithm $\mathcal{A}$ on a time budget, by alternatively learning the meta-features relative to $\mathcal{A}$, and selecting the datasets most diverse according to these meta-features, in the spirit of experiment design.

ETHICS STATEMENT

The approach is not concerned with privacy and confidentiality of the data.
The AutoML goal aims to reduce the computational resources needed to get the peak performance from an ML portfolio of algorithms or pipelines.

ACKNOWLEDGMENTS

We gratefully thank the anonymous reviewers for their constructive comments and suggestions.

This work, and in particular Herilalaina Rakotoarison, is fully funded by ADEME (#1782C0034) project NEXT.

This work is also supported by TAILOR, an ICT48 network funded by EU Horizon 2020 programme GA 952215.

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

# Learning Meta-Features for AutoML

## Supplementary Material

The supplementary material includes additional details on:

- The augmentation of the OpenML benchmark (Appendix A);
- The pseudo-code of the algorithm (Appendix B);
- The experimental setting, performance indicators and validation procedure (Appendix C);
- The hyper-parameter configuration space (Appendix D);
- The list of basic meta-features and baseline meta-featuresets (Appendix D);
- The details of the computational time (Appendix F);
- The insights into the AutoML problem provided by the approach: intrinsic dimensionality (Appendix G.1) and visualization of the niches of the considered ML algorithms (Appendix G.2);
- The detailed results with standard deviation on all three tasks (Appendix H).
- Pairwise comparison of METABU with baseline meta-features (Appendix I).
- Sensitivity analysis of dimension $d$ (Appendix J).
- Performance curves from Task 2 (Appendix K).

## A  THE AUGMENTED OPENML BENCHMARK SUITE

The visualisation of the augmented benchmark (Fig. 6, projected using tSNE (van der Maaten & Hinton, 2008) on the basic representation space), shows that the datasets built by bootstrapping of some initial dataset $E$ form a cluster close to $E$ (as could have been expected as the manually designed meta-features are stable under small stochastic variations), and separated from the clusters generated from other datasets, suggesting that the initial benchmark suite only sparsely paves the basic meta-feature space. Complementary experiments (omitted) with perplexity ranging in [5, 10, 15, 25, 30, 40, 50] show that clusters generated by augmentation of different OpenML datasets keep staying far apart from one another.

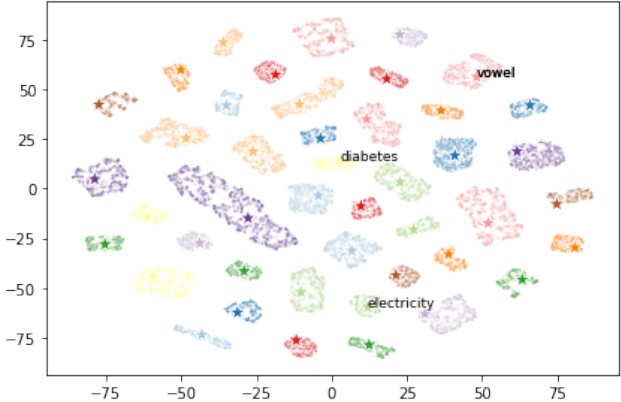

Figure 6: 2-D tSNE Visualisation of the OpenML datasets in basic representation space (legended with a $\star$'s) + their boostrapped augmentations. Only few dataset names are present for the sake of readability.

## B   PSEUDO CODE OF METABU

The learning procedure of METABU is described in Alg. 1, detailing the description presented in Section 3 of the main paper. The density on the hyper-parameter space, used to sample hyper-parameter configurations for a given dataset depending on the considered meta-features (involved in Tasks 2 and 3) is presented in Alg. 2.

---

**Algorithm 1:** Learning METABU meta-features

**Data:** Set of $n$ training datasets, each represented with its **basic representation** (meta-feature vector) $x_i$ and its **target representation** (set of top 20 hyper-parameters) $\mathbf{z}_i$ for $i = 1 \ldots n$.

**Result:** Embedding layer $\psi^*$

```
// Build projected target representation
```
1   $C_{i,j} \leftarrow d_{\mathcal{W}}^2(\mathbf{z}_i, \mathbf{z}_j)$ for $i = 1 \ldots n, j = 1 \ldots n$; /* Pairwise Wass.  Dist.     */
2   Estimate intrinsic dimension $d$ from matrix $C$ using (Facco et al., 2017);
3   $\mathbf{u} \leftarrow \text{MDS}(C, d)$ ;                               /* Multidimensional Scaling */
```
// Learn ψ
```
4   $\psi \leftarrow \text{Linear}(135, d)$ ;                              /* 135 basic meta-features.  */
5   $\mathbf{x} \leftarrow \frac{1}{p} \sum_{i=1}^{n} \delta_{x_i}$;
6   $\mathcal{L} \leftarrow \text{FGW}$ as defined in Eq. 1;
7   $\psi^* \leftarrow \text{ADAM}(\mathcal{L}, \psi_{\sharp}\mathbf{x}, \mathbf{u})$;

---

**Algorithm 2:** Fit_density

**Data:** Set of $n$ training datasets, each represented with its meta-feature vector $x_i$ and its set of top 20 hyper-parameters $\mathbf{\Theta}_i$ for $i = 1 \ldots n$. Test dataset represented with its meta-feature vector $x$.

**Result:** Distribution of configurations $\mathbf{z}$.

1   Order training datasets such that: $||x - x_\ell|| < ||x - x_{\ell+1}||$ for $\ell = 1 \ldots n$;
2   $\mathbf{z} = \frac{1}{Z} \sum_{i=\ell}^{10} exp(-\ell) \sum_{\theta \in \mathbf{\Theta}_\ell} \theta$;

---

## C    MEASURING PERFORMANCE INDICATORS

As said, the OpenML benchmark includes 72 datasets, with only 64 of them having a target representation. The other 8 datasets are too heavy (e.g. ImageNet) to launch the many runs required to estimate their target representation.

For Task 1, the performance indicator is measured along a Leave-One-Out procedure, with 64 folds: in each fold, all datasets but one are used to train the METABU meta-features; the NDCG@k is measured on the remaining dataset. Eventually, the NDCG@k are averaged over all 64 folds.

For Tasks 2 and 3, the performance indicator is likewise measured using a Leave-One-Out procedure with 64 folds. The difference is that besides the remaining dataset, the 8 datasets with no target representation at all are also used as test datasets.

In Tasks 2 and 3, the performance associated with a hyper-parameter configuration for a dataset is computed after training the model on 1 CPU with time budget of 15 mn, with memory less than 8Gb, using the train/validation/test splits given by OpenML; the validation score is estimated using a 5-CV strategy.

In Task 2, for each test dataset, and for each meta-feature set $mf$ in {METABU, AutoSkLearn, Landmark, SCOT}:

- The distribution

$$\hat{\mathbf{z}}_{mf} = \frac{1}{Z} \sum_{\ell=1}^{10} exp(-\ell)\mathbf{z}_\ell$$

  is defined, with $\mathbf{z}_\ell$ the target representation of the $\ell$-th neighbor of the considered (test) dataset, among the training datasets, according to the Euclidean distance based on their $mf$ representations.
- For $1 \leq t \leq T$, a hyper-parameter configuration is independently drawn from $\hat{\mathbf{z}}_{mf}$, and a model is learned using this configuration;
- The performance of this model is measured on a validation dataset;

- The model with best validation performance up to iteration $t$ is retained for each meta-feature $mf$ and its performance on the test dataset is computed;
- The rank $r(t, mf)$ is determined by comparing the performance on the test set, of the models retained for each meta-feature type.
- The performance curve reports $r(t, mf)$, averaged over test datasets.

In Task 3, the meta-features are used to initialize the performance model:

- In AutoSkLearn, the performance model for AutoSkLearn is initialized as follows. The best configurations for the top-10 neighbors of the current dataset are retained and run on the current dataset; their performance is used to initialize the Bayesian Optimisation search using the SMAC BO implementation Hutter et al. (2011). These top-10 neighbors are computed using the Euclidean distance on the meta-feature set. Note that AutoSkLearn meta-features have been crafted to achieve automatic configuration selection in the context of the AutoSkLearn pipeline (Pedregosa et al., 2011), thus constituting a most strong baseline on Task 3.
- For PMF, the best configurations for the top-5 neighbors of the current dataset are likewise selected; their performance is computed to fill the row of the collaborative matrix associated to the current dataset, and determine the latent representation of the current dataset. The probabilistic model learned from the matrix is used to select further hyper-parameter configurations; their performances are computed and used to refine the latent representation of the dataset.

# D   THE HYPER-PARAMETER CONFIGURATION SPACES

The hyper-parameters used for Adaboost, Random Forest and SVM and their range are detailed in Tables 1 and 2. For AutoSkLearn, we only included the list of considered hyper-parameters; their ranges are detailed in AutoSkLearn (Feurer et al., 2015). The hyper-parameter space used in PMF is the same as in AutoSkLearn. The METABU implementation uses the ConfigSpace library (Lindauer et al., 2019) to manage the hyper-parameters.

| Classifier | HP | Range |
|---|---|---|
| Adaboost | imputation | mean, median, most frequent |
| | n_estimator | $[50, 500]$ |
| | algorithm | SAMME, SAMME.R |
| | max_depth | $[1, 10]$ |
| RF | imputation | mean, median, most frequent |
| | criterion | gini, entropy |
| | max_features | $]0, 1]$ |
| | min_samples_split | $[2, 20]$ |
| | min_samples_leaf | $[1, 20]$ |
| | bootstrap | True, False |
| SVM | imputation | mean, median, most frequent |
| | C | $[0.03125, 32768]$ |
| | kernal | rbf, poly, sigmoid |
| | degree | $[1, 5]$ |
| | gamma | $[3.0517578125 \times 10^{-5}, 8]$ |
| | coef0 | $[-1, 1]$ |
| | shrinking | True, False |
| | tol | $[10^{-5}, 10^{-1}]$ |

Table 1: Hyper-parameter ranges of Adaboost, Random Forest and SVM

| Methods | Parameters |
|---|---|
| balancing | strategy |
| adaboost | learning_rate, max_depth, n_estimators |
| bernoulli_nb | fit_prior |
| decision_tree | max_depth_factor, max_features, max_leaf_nodes, min_impurity_decrease, min_samples_leaf, min_samples_split, min_weight_fraction_leaf |
| extra_trees | criterion, max_depth, max_features, max_leaf_nodes, min_impurity_decrease, min_samples_leaf, min_samples_split, min_weight_fraction_leaf |
| gradient_boosting | l2_regularization, learning_rate, loss, max_bins, max_depth, max_leaf_nodes, min_samples_leaf, scoring, tol, n_iter_no_change, validation_fraction |
| k_nearest_neighbors | p, weights |
| lda | tol, shrinkage_factor |
| liblinear_svc | dual, fit_intercept, intercept_scaling, loss, multi_class, penalty, tol |
| libsvm_svc | gamma, kernel, max_iter, shrinking, tol, coef0, degree |
| mlp | alpha, batch_size, beta_1, beta_2, early_stopping, epsilon, hidden_layer_depth, learning_rate_init, n_iter_no_change, num_nodes_per_layer, shuffle, solver, tol, validation_fraction |
| multinomial_nb | fit_prior |
| passive_aggressive | average, fit_intercept, loss, tol |
| qda | reg_param |
| random_forest | criterion, max_depth, max_features, max_leaf_nodes, min_impurity_decrease, min_samples_leaf, min_samples_split, min_weight_fraction_leaf |
| sgd | average, fit_intercept, learning_rate, loss, penalty, tol, epsilon, eta0, l1_ratio, power_t |
| extra_trees_preproc_for_classification | criterion, max_depth, max_features, max_leaf_nodes, min_impurity_decrease, min_samples_leaf, min_samples_split, min_weight_fraction_leaf, n_estimators |
| fast_ica | fun, whiten, n_components |
| feature_agglomeration | linkage, n_clusters, pooling_func |
| kernel_pca | n_components, coef0, degree, gamma |
| kitchen_sinks | n_components |
| liblinear_svc_preprocessor | dual, fit_intercept, intercept_scaling, loss, multi_class, penalty, tol |
| nystroem_sampler | n_components, coef0, degree, gamma |
| pca | whiten |
| polynomial | include_bias, interaction_only |
| random_trees_embedding | max_depth, max_leaf_nodes, min_samples_leaf, min_samples_split, min_weight_fraction_leaf, n_estimators |
| select_percentile_classification | score_func |
| select_rates_classification | score_func, mode |

Table 2: List of hyper-parameters considered in AutoSkLearn pipeline.

# E LIST OF META-FEATURES

The list of meta-features used in the experiments is detailed in Tables 3 and 4. Meta-features are extracted with PyMFE (Alcobaça et al., 2020) except for AutoSkLearn, SCOT and Landmark meta-features which are computed from the AutoSkLearn library.

| Meta-features | Description | AutoSkLearn | Landmark | SCOT | Metabu |
|---|---|---|---|---|---|
| best_node | Performance of a the best single decision tree node. | | | | + |
| elite_nn | Performance of Elite Nearest Neighbor. | | | | + |
| linear_discr | Performance of the Linear Discriminant classifier. | | | | + |
| naive_bayes | Performance of the Naive Bayes classifier. | | | | + |
| one_nn | Performance of the 1-Nearest Neighbor classifier. | | | | + |
| random_node | Performance of the single decision tree node model induced by a random attribute. | | | | + |
| worst_node | Performance of the single decision tree node model induced by the worst informative attribute. | | | | + |
| one_itemset | Compute the one itemset meta-feature. | | | | + |
| two_itemset | Compute the two itemset meta-feature. | | | | + |
| c1 | Compute the entropy of class proportions. | | | | + |
| c2 | Compute the imbalance ratio. | | | | + |
| cls_coef | Clustering coefficient. | | | | + |
| density | Average density of the network. | | | | + |
| f1 | Maximum Fisher's discriminant ratio. | | | | + |
| f1v | Directional-vector maximum Fisher's discriminant ratio. | | | | + |
| f2 | Volume of the overlapping region. | | | | + |
| f3 | Compute feature maximum individual efficiency. | | | | + |
| f4 | Compute the collective feature efficiency. | | | | + |
| hubs | Hub score. | | | | + |
| l1 | Sum of error distance by linear programming. | | | | + |
| l2 | Compute the OVO subsets error rate of linear classifier. | | | | + |
| l3 | Non-Linearity of a linear classifier. | | | | + |
| lsc | Local set average cardinality. | | | | + |
| n1 | Compute the fraction of borderline points. | | | | + |
| n2 | Ratio of intra and extra class nearest neighbor distance. | | | | + |
| n3 | Error rate of the nearest neighbor classifier. | | | | + |
| n4 | Compute the non-linearity of the k-NN Classifier. | | | | + |
| t1 | Fraction of hyperspheres covering data. | | | | + |
| t2 | Compute the average number of features per dimension. | | | | + |
| t3 | Compute the average number of PCA dimensions per points. | | | | + |
| t4 | Compute the ratio of the PCA dimension to the original dimension. | | | | + |
| ch | Compute the Calinski and Harabasz index. | | | | + |
| int | Compute the INT index. | | | | + |
| nre | Compute the normalized relative entropy. | | | | + |
| pb | Compute the pearson correlation between class matching and instance distances. | | | | + |
| sc | Compute the number of clusters with size smaller than a given size. | | | | + |
| sil | Compute the mean silhouette value. | | | | + |
| vdb | Compute the Davies and Bouldin Index. | | | | + |
| vdu | Compute the Dunn Index. | | | | + |
| leaves | Compute the number of leaf nodes in the DT model. | | | | + |
| leaves_branch | Compute the size of branches in the DT model. | | | | + |
| leaves_corrob | Compute the leaves corroboration of the DT model. | | | | + |
| leaves_homo | Compute the DT model Homogeneity for every leaf node. | | | | + |
| leaves_per_class | Compute the proportion of leaves per class in DT model. | | | | + |
| nodes | Compute the number of non-leaf nodes in DT model. | | | | + |
| nodes_per_attr | Compute the ratio of nodes per number of attributes in DT model. | | | | + |
| nodes_per_inst | Compute the ratio of non-leaf nodes per number of instances in DT model. | | | | + |
| nodes_per_level | Compute the ratio of number of nodes per tree level in DT model. | | | | + |
| nodes_repeated | Compute the number of repeated nodes in DT model. | | | | + |
| tree_depth | Compute the depth of every node in the DT model. | | | | + |
| tree_imbalance | Compute the tree imbalance for each leaf node. | | | | + |
| tree_shape | Compute the tree shape for every leaf node. | | | | + |
| var_importance | Compute the features importance of the DT model for each attribute. | | | | + |
| can_cor | Compute canonical correlations of data. | | | | + |
| cor | Compute the absolute value of the correlation of distinct dataset column pairs. | | | | + |
| cov | Compute the absolute value of the covariance of distinct dataset attribute pairs. | | | | + |
| eigenvalues | Compute the eigenvalues of covariance matrix from dataset. | | | | + |
| g_mean | Compute the geometric mean of each attribute. | | | | + |
| gravity | Compute the distance between minority and majority classes center of mass. | | | | + |
| h_mean | Compute the harmonic mean of each attribute. | | | | + |
| iq_range | Compute the interquartile range (IQR) of each attribute. | | | | + |
| kurtosis | Compute the kurtosis of each attribute. | | | | + |
| lh_trace | Compute the Lawley-Hotelling trace. | | | | + |
| mad | Compute the Median Absolute Deviation (MAD) adjusted by a factor. | | | | + |
| max | Compute the maximum value from each attribute. | | | | + |
| mean | Compute the mean value of each attribute. | | | | + |
| median | Compute the median value from each attribute. | | | | + |
| min | Compute the minimum value from each attribute. | | | | + |
| nr_cor_attr | Compute the number of distinct highly correlated pair of attributes. | | | | + |
| nr_disc | Compute the number of canonical correlation between each attribute and class. | | | | + |
| nr_norm | Compute the number of attributes normally distributed based in a given method. | | | | + |
| nr_outliers | Compute the number of attributes with at least one outlier value. | | | | + |
| p_trace | Compute the Pillai's trace. | | | | + |
| range | Compute the range (max - min) of each attribute. | | | | + |

Table 3: List of meta-features, 1/2

| Meta-features | Description | AutoSkLearn | Landmark | SCOT | Metabu |
|---|---|:---:|:---:|:---:|:---:|
| roy_root | Compute the Roy's largest root. | | | | + |
| sd | Compute the standard deviation of each attribute. | | | | + |
| sd_ratio | Compute a statistical test for homogeneity of covariances. | | | | + |
| skewness | Compute the skewness for each attribute. | | | | + |
| sparsity | Compute (possibly normalized) sparsity metric for each attribute. | | | | + |
| t_mean | Compute the trimmed mean of each attribute. | | | | + |
| var | Compute the variance of each attribute. | | | | + |
| w_lambda | Compute the Wilks' Lambda value. | | | | + |
| attr_conc | Compute concentration coef. of each pair of distinct attributes. | | | | + |
| attr_ent | Compute Shannon's entropy for each predictive attribute. | | | | + |
| class_conc | Compute concentration coefficient between each attribute and class. | | | | + |
| class_ent | Compute target attribute Shannon's entropy. | | | | + |
| eq_num_attr | Compute the number of attributes equivalent for a predictive task. | | | | + |
| joint_ent | Compute the joint entropy between each attribute and class. | | | | + |
| mut_inf | Compute the mutual information between each attribute and target. | | | | + |
| ns_ratio | Compute the noisiness of attributes. | | | | + |
| cohesiveness | Compute the improved version of the weighted distance, that captures how dense or sparse is the example distribution. | | | | + |
| conceptvar | Compute the concept variation that estimates the variability of class labels among examples. | | | | + |
| impconceptvar | Compute the improved concept variation that estimates the variability of class labels among examples. | | | | + |
| wg_dist | Compute the weighted distance, that captures how dense or sparse is the example distribution. | | | | + |
| attr_to_inst | Compute the ratio between the number of attributes. | | | | + |
| cat_to_num | Compute the ratio between the number of categoric and numeric features. | | | | + |
| freq_class | Compute the relative frequency of each distinct class. | | | | + |
| inst_to_attr | Compute the ratio between the number of instances and attributes. | | | | + |
| nr_attr | Compute the total number of attributes. | | | | + |
| nr_bin | Compute the number of binary attributes. | | | | + |
| nr_cat | Compute the number of categorical attributes. | | | | + |
| nr_class | Compute the number of distinct classes. | | | | + |
| nr_inst | Compute the number of instances (rows) in the dataset. | | | | + |
| nr_num | Compute the number of numeric features. | | | | + |
| num_to_cat | Compute the number of numerical and categorical features. | | | | + |
| PCASkewnessFirstPC | Skewness of examples on the first principal component | | | | + |
| PCAKurtosisFirstPC | Kurtosis of examples on the first principal component | | | | + |
| PCAFracOfCompFor95Per | Fraction of component of an overall explained variance of 95% | | | + | + |
| Landmark1NN | Performance one nearest neighbor classifier | | | | + |
| LandmarkRandomNodeLearner | Performance of decision when considering only one feature | | | | + |
| LandmarkDecisionNodeLearner | Performance of decision when considering all features | | | | + |
| LandmarkDecisionTree | Performance of decision tree classifier | | + | | + |
| LandmarkNaiveBayes | Performance of Naive Bayes classifier | | + | | + |
| LandmarkLDA | Performance of LDA classifier | | + | | + |
| SkewnessSTD | Standard deviation of feature skewness | + | | | + |
| SkewnessMean | Mean of feature skewness | + | | | + |
| SkewnessMax | Maximum of feature skewness | + | | | + |
| SkewnessMin | Minimum of feature skewness | + | | | + |
| KurtosisSTD | Standard deviation of feature kurtosis coefficiants | + | | | + |
| KurtosisMean | Mean of feature kurtosis coefficiants | + | | | + |
| KurtosisMax | Max of feature kurtosis coefficiants | + | | | + |
| KurtosisMin | Mean of feature kurtosis coefficiants | + | | | + |
| SymbolsSum | Sum of categorical feature symbols | + | | | + |
| SymbolsSTD | Standard deviation of categorical feature symbols | + | | | + |
| SymbolsMean | Mean of categorical feature symbols | + | | | + |
| SymbolsMax | Max of categorical feature symbols | + | | | + |
| SymbolsMin | Min of categorical feature symbols | + | | | + |
| ClassProbabilitySTD | Standard deviation of class probabilities | + | | | + |
| ClassProbabilityMean | Mean of class probabilities | + | | | + |
| ClassProbabilityMax | Maximum of class probabilities | + | + | | + |
| ClassProbabilityMin | Minimum of class probabilities | + | | | + |
| InverseDatasetRatio | Inverse of dataset ratio | + | | | + |
| DatasetRatio | Dataset ratio | + | | | + |
| RatioNominalToNumerical | Ratio number of nominal to numerical features | + | | | + |
| RatioNumericalToNominal | Ratio numerical to nominal | + | | | + |
| NumberOfCategoricalFeatures | Number of categorical features | + | + | | + |
| NumberOfNumericFeatures | Number of numeric features | + | + | | + |
| NumberOfMissingValues | Number of missing values | + | | | + |
| NumberOfFeaturesWithMissingValues | Number of features with missing values | + | | | + |
| NumberOfInstancesWithMissingValues | Number of instances with missing values | + | | | + |
| NumberOfFeatures | Number of features | + | + | | + |
| NumberOfClasses | Number of classes | + | + | + | + |
| NumberOfInstances | Number of instances | + | | | + |
| LogInverseDatasetRatio | log of the inverse dataset ratio | + | | + | + |
| LogDatasetRatio | Log of dataset ratio | + | | | + |
| PercentageOfMissingValues | Percentage of missing values | + | | | + |
| PercentageOfFeaturesWithMissingValues | Percentage of features with missing values | + | | | + |
| PercentageOfInstancesWithMissingValues | Percentage of instances with missing values | + | | | + |
| LogNumberOfFeatures | Log number of features | + | | + | + |
| LogNumberOfInstances | Log number of instances | + | | | + |

Table 4: List of meta-features, 2/2

## F COMPUTATIONAL EFFORT

Fig. 7 indicates the runtime[6] for pre-processing (extracting the 135 meta-features, top row), and for training METABU (second row). The training times for learning one model is indicated for comparison (from row 3 to 5: Adaboost, RandomForest and SVM).

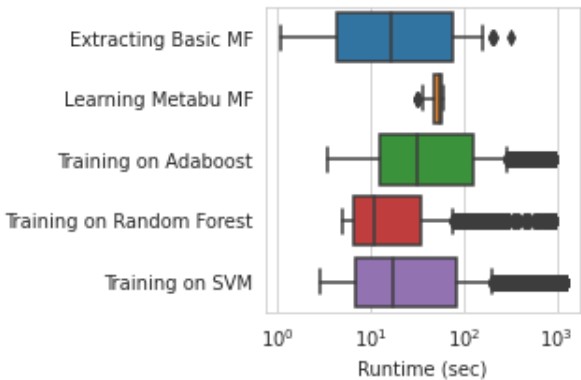

Figure 7: METABU computational effort: average runtime of the meta-feature extraction (in blue) and METABU training (in orange). The average training time of one hyper-parameter on Adaboost (green), Random Forest (red) and SVM (purple) pipelines are shown for comparison.

---

[6]On Intel(R) Xeon(R) CPU E5-2660 v2 @ 2.20GHz.

# G TOWARD UNDERSTANDING THE DATASET LANDSCAPE

## G.1 THE STABILITY OF THE INTRINSIC DIMENSION

| Dataset Ratio | 0.1 | 0.25 | 0.5 | 0.75 | 1 |
|---|---|---|---|---|---|
| Adaboost | 5.65 (2.74) | 6.81 (1.81) | 7.14 (1.44) | 6.59 (1.29) | 6.98 |
| Random Forest | 5.33 (2.17) | 7.14 (2.18) | 7.44 (1.28) | 8.48 (1.56) | 8.49 |
| SVM | 8.56 (2.54) | 11.54 (2.71) | 12.83 (3.16) | 13.99 (2.40) | 14.41 |
| AutoSkLearn | 5.17 (2.08) | 4.47 (1.26) | 4.98 (0.95) | 5.34 (1.06) | 5.51 |

Table 5: Intrinsic dimension of the dataset space w.r.t. ML algorithms Adaboost, RandomForest, SVM and AutoSkLearn, depending on the fraction of datasets considered in OpenML

In Table 5, we investigate how the intrinsic dimension varies when considering various numbers of datasets in OpenML. It is observed that the intrinsic dimension tends to increase with the number of considered datasets, particularly so for SVM and to a lesser extent for RandomForest. This suggests that the hyper-parameter configurations investigated in the OpenML benchmark for these algorithms do not sufficiently sample the (good regions of the) configuration spaces.

## G.2 INTERPRETATION: IMPACT OF THE HC META-FEATURES ON THE PERFORMANCE OF THE LEARNING ALGORITHM

METABU meta-features are built from the initial meta-features using the trained linear mapping $\psi$, depending on the current learning algorithm $\mathcal{A}$. Accordingly, the importance of the initial, *humanly defined and interpretable* meta-features w.r.t. $\mathcal{A}$ can be estimated, shedding some light on which specifics of a dataset matter in order to give a good/bad performance with $\mathcal{A}$.

The importance of a meta-feature w.r.t. $\mathcal{A}$ is estimated as follows. Let $U = \{u_{i,j}\}$ denote the matrix made of the multi-dimensional scaling representation of the target representation over all datasets (section 3.2). Let $v$ denote the first principal component of $U$ and let $j^*$ be the index of the $H$ column most contributing to $v$ ($j^* = argmax_j|\langle v, h_{.,j}\rangle|$). Then the importance $i_{\mathcal{A}}(k)$ of the $k$-th initial meta-feature for the $\mathcal{A}$ algorithm is defined as the absolute value of $\psi_{j^*,k}$, that is, the weight of the $k$-th initial meta-feature to build the most important METABU meta-feature.

This estimate is used to visually appreciate the meta-feature importance w.r.t. two learning algorithms $\mathcal{A}$ and $\mathcal{B}$, by depicting each $k$-th meta-feature in the 2D plane as the point with coordinate $(i_{\mathcal{A}}(k), i_{\mathcal{B}}(k))$. Intuitively, meta-features on the diagonal have the same importance for both algorithms. Meta-features far from the diagonal are much more important for one algorithm than for the other. The visualization of the meta-feature importance w.r.t. AutoSkLearn and Random Forest is displayed in Fig. 8, left. Meta-features such as KurtosisMin, LogNumberOfInstances, InverseDatasetRatio − all retained as AutoSkLearn meta-features − are critical for AutoSkLearn whereas they have no impact for RandomForest. Inversely, some features like "pb" (average Pearson correlation between class and features) matter significantly more for RandomForest than for AutoSkLearn.

Likewise, the meta-feature importance w.r.t. Support Vector Machines and Random Forest is displayed in Fig. 8, right. The skewness features (mean and std deviation over all attributes) matter significantly more for Support Vector Machines than for RandomForest. In retrospect, there is little surprise that the meta-features related to the potential difficulties of inverting the Gram matrix matter for SVM.

Overall, the impact of some meta-features for some learning algorithms is intuitive; it confirms the practitioner expertise, which is comforting.

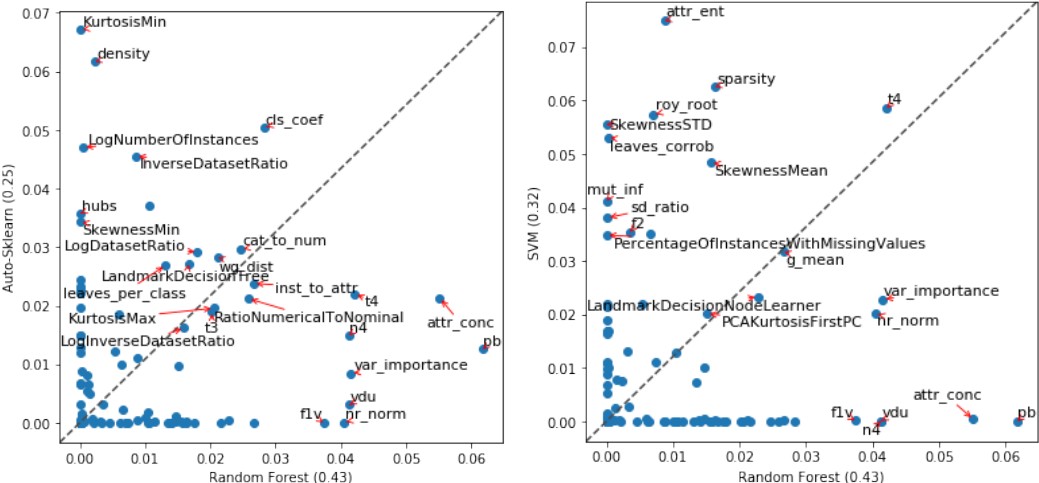

Figure 8: Comparative importance of meta-features for RandomForest Vs AutoSkLearn (left) and SVM (right). The specific AutoSkLearn meta-features are indicated as their name begins with a capital letter.

# H   DETAILED RESULTS

Detailed results of Task 2 are presented in Table 6 for Random Forest, Table 7 for Adaboost and Table 8 for SVM. We consider the Mann Whitney Wilcoxon test to assess the significance of the rankings.

| OpenML Task ID | METABU MF | AutoSkLearn MF | Landmark MF | SCOT MF | Random1x |
|---|---|---|---|---|---|
| Average Rank | **2.50** | 2.97 | 2.70 | 2.64 | 4.17 |
| 3 | 0.993 ±0.000* | 0.993 ± 0.001 | 0.993 ±0.000* | 0.993 ± 0.000 | 0.993 ± 0.001 |
| 6 | 0.965 ±0.001* | 0.964 ± 0.001 | 0.958 ± 0.007 | 0.964 ±0.002* | 0.948 ± 0.007 |
| 11 | 0.655 ±0.002* | 0.657 ± 0.002 | 0.658 ±0.001* | 0.656 ± 0.001 | 0.657 ± 0.002 |
| 12 | 0.964 ±0.002* | 0.961 ± 0.001 | 0.965 ±0.001* | 0.963 ± 0.003 | 0.961 ± 0.004 |
| 14 | 0.820 ±0.005* | 0.812 ± 0.004 | 0.813 ± 0.005 | 0.814 ±0.003* | 0.808 ± 0.005 |
| 15 | 0.983 ±0.004* | 0.982 ± 0.005 | 0.983 ±0.001* | 0.981 ± 0.005 | 0.983 ± 0.004 |
| 16 | 0.955 ±0.007* | 0.951 ± 0.010 | 0.958 ± 0.004 | 0.959 ±0.003* | 0.950 ± 0.005 |
| 18 | 0.675 ±0.002* | 0.673 ± 0.005 | 0.676 ± 0.002 | 0.678 ± 0.001 | 0.679 ±0.007* |
| 22 | 0.765 ±0.001* | 0.761 ± 0.004 | 0.760 ± 0.005 | 0.769 ±0.003* | 0.748 ± 0.014 |
| 23 | 0.535 ± 0.004 | 0.535 ± 0.004 | 0.541 ± 0.005 | 0.542 ± 0.008 | **0.552 ± 0.006** |
| 28 | 0.983 ±0.001* | 0.982 ± 0.001 | 0.983 ±0.001* | 0.983 ±0.001* | 0.978 ± 0.001 |
| 29 | 0.884 ±0.006* | 0.878 ± 0.004 | 0.882 ±0.007* | 0.878 ± 0.005 | 0.882 ± 0.004 |
| 31 | 0.709 ±0.003* | 0.725 ±0.013* | 0.716 ± 0.003 | 0.707 ± 0.010 | 0.713 ± 0.007 |
| 32 | 0.993 ±0.001* | 0.992 ± 0.001 | 0.992 ± 0.001 | 0.994 ±0.000* | 0.989 ± 0.000 |
| 37 | 0.811 ±0.003* | 0.810 ± 0.006 | 0.812 ±0.005* | 0.810 ± 0.011 | 0.808 ± 0.007 |
| 43 | 0.913 ±0.002* | 0.914 ± 0.002 | 0.915 ± 0.002 | 0.917 ±0.001* | 0.908 ± 0.003 |
| 45 | 0.946 ± 0.001 | 0.946 ± 0.002 | 0.947 ± 0.002 | **0.953 ± 0.004** | 0.944 ± 0.003 |
| 49 | 0.962 ±0.002* | 0.965 ±0.002* | 0.964 ± 0.000 | 0.963 ± 0.002 | 0.957 ± 0.005 |
| 53 | 0.780 ±0.009* | 0.777 ± 0.001 | 0.786 ±0.006* | 0.767 ± 0.012 | 0.768 ± 0.006 |
| 219 | 0.923 ±0.002* | 0.919 ± 0.009 | 0.918 ± 0.004 | 0.923 ±0.003* | 0.913 ± 0.001 |
| 2074 | 0.891 ±0.001* | 0.889 ± 0.003 | 0.889 ±0.002* | 0.888 ± 0.003 | 0.880 ± 0.003 |
| 2079 | 0.638 ±0.007* | 0.628 ± 0.010 | 0.647 ±0.005* | 0.639 ± 0.007 | 0.621 ± 0.009 |
| 3021 | 0.949 ±0.003* | 0.943 ± 0.004 | 0.949 ±0.003* | 0.945 ± 0.005 | 0.933 ± 0.005 |
| 3022 | 0.962 ±0.003* | 0.959 ±0.010* | 0.945 ± 0.017 | 0.927 ± 0.014 | 0.906 ± 0.025 |
| 3549 | 0.951 ± 0.015 | **0.984 ± 0.002** | 0.967 ± 0.023 | 0.977 ± 0.007 | 0.957 ± 0.023 |
| 3560 | 0.253 ±0.006* | 0.253 ±0.021* | 0.248 ± 0.017 | 0.250 ± 0.023 | 0.241 ± 0.013 |
| 3902 | 0.756 ±0.002* | 0.754 ±0.009* | 0.734 ± 0.009 | 0.743 ± 0.020 | 0.753 ± 0.007 |
| 3903 | 0.551 ±0.013* | 0.554 ± 0.003 | 0.555 ± 0.008 | 0.557 ±0.010* | 0.549 ± 0.006 |
| 3904 | 0.602 ± 0.001 | 0.602 ± 0.003 | 0.600 ± 0.002 | **0.606 ± 0.001** | 0.594 ± 0.002 |
| 3913 | 0.619 ± 0.006 | 0.616 ± 0.016 | **0.634 ± 0.013** | 0.633 ± 0.001 | 0.609 ± 0.021 |
| 3917 | 0.667 ±0.015* | 0.677 ±0.009* | 0.669 ± 0.002 | 0.672 ± 0.005 | 0.659 ± 0.004 |
| 3918 | 0.655 ±0.003* | 0.661 ±0.008* | 0.651 ± 0.008 | 0.649 ± 0.006 | 0.652 ± 0.009 |
| 7592 | 0.778 ±0.002* | 0.781 ±0.001* | 0.777 ± 0.003 | 0.778 ± 0.002 | 0.777 ± 0.002 |
| 9910 | 0.807 ±0.002* | 0.809 ±0.002* | 0.807 ± 0.004 | 0.805 ± 0.002 | 0.797 ± 0.007 |
| 9946 | 0.953 ±0.006* | 0.951 ± 0.007 | 0.957 ± 0.008 | 0.962 ±0.011* | 0.941 ± 0.010 |
| 9952 | 0.890 ±0.001* | 0.891 ±0.001* | 0.882 ± 0.007 | 0.880 ± 0.006 | 0.878 ± 0.003 |
| 9957 | 0.866 ±0.007* | 0.864 ± 0.008 | 0.858 ± 0.002 | 0.872 ±0.002* | 0.868 ± 0.005 |
| 9960 | 0.994 ±0.000* | 0.993 ± 0.000 | 0.994 ±0.000* | 0.993 ± 0.000 | 0.993 ± 0.000 |
| 9964 | 0.927 ±0.007* | 0.915 ±0.020* | 0.912 ± 0.014 | 0.914 ± 0.005 | 0.891 ± 0.015 |
| 9971 | 0.563 ± 0.018 | **0.587 ± 0.005** | 0.584 ± 0.032 | 0.560 ± 0.020 | 0.566 ± 0.006 |
| 9976 | 0.845 ±0.007* | 0.846 ±0.004* | 0.841 ± 0.010 | 0.840 ± 0.005 | 0.842 ± 0.006 |
| 9977 | 0.961 ±0.000* | 0.960 ± 0.000 | 0.961 ±0.000* | 0.961 ± 0.001 | 0.960 ± 0.001 |
| 9978 | 0.672 ±0.007* | 0.680 ±0.006* | 0.671 ± 0.003 | 0.677 ± 0.009 | 0.670 ± 0.004 |
| 9981 | 0.926 ± 0.002 | 0.936 ± 0.011 | **0.947 ± 0.019** | 0.943 ± 0.030 | 0.927 ± 0.022 |
| 9985 | 0.475 ±0.007* | 0.478 ± 0.004 | 0.475 ± 0.002 | 0.479 ±0.004* | 0.467 ± 0.007 |
| 10093 | 0.983 ±0.001* | 0.984 ± 0.001 | 0.988 ± 0.004 | 0.988 ±0.007* | 0.987 ± 0.003 |
| 10101 | 0.621 ±0.004* | 0.611 ± 0.005 | 0.621 ±0.005* | 0.616 ± 0.005 | 0.614 ± 0.003 |
| 14952 | 0.965 ±0.000* | 0.963 ± 0.002 | 0.965 ±0.001* | 0.963 ± 0.001 | 0.956 ± 0.004 |
| 14954 | 0.844 ±0.022* | 0.835 ± 0.023 | 0.853 ±0.009* | 0.833 ± 0.004 | 0.799 ± 0.013 |
| 14965 | 0.711 ±0.001* | 0.712 ±0.002* | 0.710 ± 0.004 | 0.710 ± 0.003 | 0.709 ± 0.002 |
| 14969 | 0.597 ±0.005* | 0.588 ± 0.007 | 0.591 ± 0.005 | 0.595 ±0.002* | 0.575 ± 0.008 |
| 125920 | 0.598 ±0.010* | 0.597 ± 0.009 | 0.600 ± 0.021 | 0.601 ±0.005* | 0.589 ± 0.010 |
| 125922 | 0.976 ±0.002* | 0.976 ± 0.002 | 0.976 ±0.001* | 0.973 ± 0.005 | 0.968 ± 0.004 |
| 146195 | 0.642 ±0.002* | 0.638 ± 0.004 | 0.644 ±0.002* | 0.642 ± 0.002 | 0.622 ± 0.005 |
| 146800 | 0.971 ± 0.013 | 0.980 ± 0.009 | 0.974 ± 0.006 | **0.986 ± 0.002** | 0.962 ± 0.010 |
| 146817 | 0.825 ±0.004* | 0.817 ± 0.009 | 0.826 ±0.007* | 0.824 ± 0.012 | 0.812 ± 0.004 |
| 146819 | 0.861 ±0.014* | 0.859 ± 0.010 | 0.861 ± 0.015 | 0.870 ±0.002* | 0.869 ± 0.005 |
| 146820 | 0.863 ±0.004* | 0.855 ± 0.014 | 0.851 ± 0.018 | 0.831 ± 0.005 | 0.855 ±0.010* |
| 146821 | 0.971 ±0.001* | 0.972 ±0.001* | 0.969 ± 0.002 | 0.970 ± 0.001 | 0.971 ± 0.004 |
| 146822 | 0.934 ±0.001* | 0.932 ± 0.005 | 0.930 ± 0.006 | 0.934 ±0.001* | 0.931 ± 0.004 |
| 146824 | 0.968 ±0.003* | 0.969 ± 0.001 | 0.968 ± 0.003 | 0.969 ±0.002* | 0.960 ± 0.007 |
| 146825 | 0.294 ±0.509* | 0.582 ± 0.504 | 0.293 ± 0.507 | 0.872 ±0.006* | 0.869 ± 0.003 |
| 167119 | 0.767 ±0.001* | 0.764 ± 0.003 | 0.762 ± 0.003 | 0.765 ±0.002* | 0.765 ± 0.000 |
| 167121 | 0.275 ±0.476* | 0.000 ± 0.000 | 0.582 ±0.505* | 0.000 ± 0.000 | 0.274 ± 0.475 |
| 167125 | 0.921 ±0.000* | 0.922 ± 0.002 | 0.924 ±0.003* | 0.920 ± 0.001 | 0.899 ± 0.007 |
| 167140 | 0.930 ± 0.004 | 0.935 ± 0.001 | 0.933 ± 0.002 | **0.938 ± 0.001** | 0.924 ± 0.003 |
| 167141 | 0.834 ±0.002* | 0.829 ± 0.005 | 0.833 ± 0.002 | 0.837 ±0.003* | 0.832 ± 0.001 |

Table 6: Comparative learning performances on OpenML datasets over sampling 30 configurations of the **Random Forest** pipeline. Performances that are statistically significant compared to the second best are in bold. Statistically comparable performances are indicated with (*). Pairwise comparison and p-value along the iterations are presented in Fig. 9.

| OpenML Task ID | METABU MF | AutoSkLearnMF | Landmark MF | SCOT MF | Random1x |
|---|---|---|---|---|---|
| Average Rank | **2.48** | 2.96 | 2.89 | 2.85 | 3.80 |
| 3 | 0.995 ±0.001* | 0.994 ± 0.000 | 0.996 ±0.001* | 0.994 ± 0.002 | 0.996 ± 0.001 |
| 6 | 0.970 ± 0.001 | 0.969 ± 0.002 | 0.967 ± 0.002 | **0.972 ± 0.001** | 0.967 ± 0.005 |
| 11 | 0.928 ±0.083* | 0.891 ± 0.071 | 0.914 ± 0.069 | 0.973 ±0.017* | 0.920 ± 0.093 |
| 12 | 0.977 ±0.001* | 0.977 ±0.001* | 0.976 ± 0.002 | 0.977 ± 0.001 | 0.975 ± 0.002 |
| 14 | 0.827 ±0.007* | 0.829 ± 0.004 | 0.824 ± 0.006 | 0.825 ± 0.002 | 0.829 ±0.004* |
| 15 | 0.964 ±0.004* | 0.962 ± 0.006 | 0.969 ±0.004* | 0.967 ± 0.005 | 0.967 ± 0.006 |
| 16 | 0.963 ±0.001* | 0.966 ± 0.001 | 0.967 ±0.004* | 0.964 ± 0.004 | 0.961 ± 0.001 |
| 18 | 0.691 ±0.015* | 0.674 ± 0.003 | 0.695 ±0.014* | 0.689 ± 0.012 | 0.680 ± 0.018 |
| 22 | 0.796 ±0.002* | 0.800 ± 0.004 | 0.801 ±0.006* | 0.794 ± 0.014 | 0.791 ± 0.005 |
| 23 | 0.572 ±0.009* | 0.579 ± 0.011 | 0.582 ±0.015* | 0.575 ± 0.004 | 0.581 ± 0.009 |
| 28 | 0.988 ±0.001* | 0.988 ± 0.001 | 0.987 ± 0.001 | 0.989 ±0.001* | 0.987 ± 0.001 |
| 29 | 0.865 ± 0.006 | 0.866 ± 0.008 | 0.875 ± 0.009 | **0.882 ± 0.006** | 0.845 ± 0.015 |
| 31 | 0.739 ±0.011* | 0.726 ± 0.007 | 0.734 ± 0.011 | 0.739 ± 0.019 | 0.740 ±0.014* |
| 32 | 0.995 ±0.001* | 0.997 ±0.000* | 0.996 ± 0.000 | 0.996 ± 0.001 | 0.994 ± 0.001 |
| 37 | 0.790 ±0.005* | 0.794 ±0.018* | 0.771 ± 0.008 | 0.782 ± 0.010 | 0.784 ± 0.002 |
| 43 | 0.936 ±0.001* | 0.933 ± 0.003 | 0.934 ±0.003* | 0.932 ± 0.004 | 0.933 ± 0.002 |
| 45 | 0.961 ±0.003* | 0.951 ± 0.005 | 0.957 ±0.005* | 0.954 ± 0.004 | 0.951 ± 0.003 |
| 49 | 0.993 ± 0.002 | 0.995 ± 0.002 | 0.997 ± 0.003 | 0.987 ± 0.012 | **0.998 ± 0.002** |
| 53 | 0.808 ±0.005* | 0.783 ± 0.027 | 0.804 ±0.003* | 0.801 ± 0.007 | 0.801 ± 0.012 |
| 219 | **0.938 ± 0.001** | 0.937 ± 0.000 | 0.931 ± 0.006 | 0.933 ± 0.003 | 0.924 ± 0.001 |
| 2074 | 0.903 ±0.002* | 0.902 ±0.001* | 0.901 ± 0.001 | 0.901 ± 0.002 | 0.901 ± 0.001 |
| 2079 | 0.657 ±0.011* | 0.649 ±0.006* | 0.648 ± 0.010 | 0.641 ± 0.007 | 0.627 ± 0.002 |
| 3021 | 0.955 ±0.005* | 0.956 ±0.004* | 0.951 ± 0.003 | 0.955 ± 0.002 | 0.953 ± 0.002 |
| 3022 | 0.952 ± 0.020 | 0.969 ± 0.001 | 0.966 ± 0.005 | **0.972 ± 0.002** | 0.960 ± 0.008 |
| 3549 | 0.988 ±0.002* | 0.988 ±0.002* | 0.988 ± 0.001 | 0.986 ± 0.001 | 0.985 ± 0.003 |
| 3560 | 0.255 ±0.017* | 0.241 ± 0.001 | 0.258 ± 0.020 | 0.262 ±0.012* | 0.253 ± 0.002 |
| 3902 | 0.762 ±0.004* | 0.769 ±0.015* | 0.754 ± 0.024 | 0.755 ± 0.012 | 0.760 ± 0.014 |
| 3903 | 0.604 ±0.027* | 0.581 ± 0.010 | 0.575 ± 0.016 | 0.574 ± 0.015 | 0.598 ±0.015* |
| 3904 | 0.615 ±0.001* | 0.608 ± 0.006 | 0.612 ± 0.003 | 0.613 ±0.002* | 0.613 ± 0.002 |
| 3913 | 0.665 ±0.037* | 0.661 ±0.018* | 0.656 ± 0.011 | 0.649 ± 0.006 | 0.659 ± 0.019 |
| 3917 | 0.681 ±0.007* | 0.682 ± 0.003 | 0.682 ± 0.008 | 0.694 ±0.015* | 0.681 ± 0.009 |
| 3918 | 0.683 ±0.019* | 0.696 ±0.008* | 0.688 ± 0.019 | 0.675 ± 0.018 | 0.685 ± 0.011 |
| 7592 | 0.798 ±0.005* | 0.797 ± 0.000 | 0.796 ± 0.000 | 0.799 ±0.004* | 0.798 ± 0.001 |
| 9910 | 0.798 ±0.001* | 0.796 ± 0.002 | 0.797 ±0.004* | 0.797 ± 0.003 | 0.793 ± 0.005 |
| 9946 | 0.977 ±0.010* | 0.986 ± 0.003 | 0.993 ±0.007* | 0.987 ± 0.011 | 0.990 ± 0.005 |
| 9952 | 0.899 ±0.004* | 0.899 ± 0.002 | 0.900 ± 0.002 | 0.900 ±0.002* | 0.896 ± 0.001 |
| 9957 | 0.871 ±0.005* | 0.871 ± 0.010 | 0.867 ± 0.006 | 0.875 ±0.007* | 0.868 ± 0.003 |
| 9960 | 0.997 ±0.001* | 0.997 ± 0.001 | 0.997 ± 0.001 | 0.997 ±0.001* | 0.996 ± 0.002 |
| 9964 | 0.932 ± 0.006 | **0.943 ± 0.003** | 0.940 ± 0.005 | 0.937 ± 0.008 | 0.928 ± 0.005 |
| 9971 | 0.563 ±0.033* | 0.573 ± 0.042 | 0.576 ±0.010* | 0.558 ± 0.007 | 0.565 ± 0.027 |
| 9976 | 0.846 ±0.003* | 0.834 ± 0.004 | 0.844 ±0.008* | 0.830 ± 0.012 | 0.833 ± 0.012 |
| 9977 | 0.964 ± 0.002 | 0.966 ± 0.002 | 0.964 ± 0.001 | 0.967 ± 0.001 | **0.967 ± 0.001** |
| 9978 | 0.699 ±0.026* | 0.683 ± 0.008 | 0.696 ±0.021* | 0.672 ± 0.016 | 0.692 ± 0.022 |
| 9981 | 0.885 ±0.011* | 0.890 ± 0.004 | 0.888 ± 0.004 | 0.894 ±0.002* | 0.881 ± 0.003 |
| 9985 | 0.475 ±0.005* | 0.473 ± 0.005 | 0.475 ±0.001* | 0.475 ± 0.002 | 0.473 ± 0.007 |
| 10093 | 0.997 ±0.003* | 0.994 ±0.001* | 0.991 ± 0.003 | 0.993 ± 0.004 | 0.994 ± 0.005 |
| 10101 | 0.619 ±0.011* | 0.615 ± 0.000 | 0.621 ± 0.002 | 0.626 ±0.013* | 0.619 ± 0.009 |
| 14952 | 0.964 ±0.002* | 0.963 ± 0.001 | 0.963 ± 0.002 | 0.963 ± 0.002 | 0.964 ±0.001* |
| 14954 | 0.869 ±0.007* | 0.881 ±0.018* | 0.872 ± 0.003 | 0.868 ± 0.002 | 0.863 ± 0.010 |
| 14965 | 0.711 ± 0.004 | 0.715 ± 0.002 | 0.714 ± 0.003 | **0.716 ± 0.002** | 0.714 ± 0.001 |
| 14969 | 0.617 ±0.004* | 0.606 ± 0.015 | 0.610 ± 0.008 | 0.610 ±0.004* | 0.607 ± 0.012 |
| 125920 | 0.572 ±0.019* | 0.569 ± 0.015 | 0.569 ±0.015* | 0.565 ± 0.005 | 0.555 ± 0.013 |
| 125922 | 0.992 ±0.001* | 0.992 ±0.000* | 0.991 ± 0.000 | 0.992 ± 0.001 | 0.989 ± 0.000 |
| 146800 | 0.996 ±0.002* | 0.996 ± 0.003 | 0.998 ±0.001* | 0.997 ± 0.001 | 0.990 ± 0.004 |
| 146817 | 0.817 ±0.008* | 0.812 ± 0.005 | 0.810 ± 0.014 | 0.821 ±0.007* | 0.805 ± 0.008 |
| 146819 | 0.797 ±0.029* | 0.766 ± 0.039 | 0.782 ± 0.024 | 0.817 ±0.025* | 0.806 ± 0.020 |
| 146820 | 0.859 ±0.004* | 0.859 ± 0.008 | 0.849 ± 0.006 | 0.855 ± 0.016 | 0.860 ±0.002* |
| 146821 | 0.980 ±0.015* | 0.974 ± 0.003 | 0.979 ± 0.009 | 0.981 ±0.004* | 0.970 ± 0.010 |
| 146822 | 0.943 ± 0.000 | 0.942 ± 0.003 | **0.945 ± 0.000** | 0.943 ± 0.004 | 0.941 ± 0.004 |
| 146824 | 0.977 ±0.001* | 0.976 ±0.002* | 0.976 ± 0.002 | 0.974 ± 0.001 | 0.972 ± 0.003 |
| 167119 | 0.818 ±0.002* | 0.816 ± 0.003 | 0.817 ±0.001* | 0.815 ± 0.003 | 0.815 ± 0.001 |
| 167125 | 0.914 ±0.002* | 0.914 ±0.000* | 0.910 ± 0.002 | 0.911 ± 0.002 | 0.912 ± 0.002 |
| 167140 | 0.954 ±0.005* | 0.954 ±0.003* | 0.953 ± 0.002 | 0.952 ± 0.003 | 0.948 ± 0.002 |
| 167141 | 0.824 ±0.004* | 0.823 ± 0.002 | 0.825 ± 0.003 | 0.826 ±0.001* | 0.822 ± 0.003 |

Table 7: Comparative learning performances on OpenML datasets over sampling 30 configurations of the **Adaboost** pipeline. Performances that are statistically significant compared to the second best are in bold. Statistically comparable performances are indicated with (*). Pairwise comparisons and the associated p-value along the iterations are reported in Fig. 10.

| OpenML Task ID | METABU MF | AutoSkLearn MF | Landmark MF | SCOT MF | Random1x |
|---|---|---|---|---|---|
| Average Rank | **2.34** | 2.91 | 2.97 | 3.27 | 3.48 |
| 3 | 0.995 ±0.001* | 0.993 ± 0.004 | 0.994 ±0.003* | 0.982 ± 0.015 | 0.988 ± 0.001 |
| 6 | 0.827 ±0.253* | 0.969 ± 0.008 | 0.973 ±0.000* | 0.967 ± 0.010 | 0.937 ± 0.000 |
| 11 | 0.967 ±0.045* | 0.929 ± 0.060 | 0.897 ± 0.104 | 0.996 ±0.007* | 0.958 ± 0.015 |
| 12 | 0.975 ±0.007* | 0.980 ± 0.001 | 0.982 ± 0.002 | 0.984 ±0.007* | 0.936 ± 0.018 |
| 14 | 0.857 ±0.029* | 0.843 ±0.010* | 0.830 ± 0.018 | 0.824 ± 0.031 | 0.839 ± 0.014 |
| 15 | 0.980 ±0.009* | 0.983 ±0.005* | 0.980 ± 0.004 | 0.982 ± 0.006 | 0.953 ± 0.002 |
| 16 | 0.981 ±0.004* | 0.979 ± 0.002 | 0.981 ±0.005* | 0.981 ± 0.003 | 0.976 ± 0.011 |
| 18 | 0.728 ±0.013* | 0.721 ± 0.025 | 0.736 ±0.010* | 0.731 ± 0.013 | 0.717 ± 0.014 |
| 22 | 0.836 ±0.016* | 0.806 ± 0.001 | 0.818 ± 0.031 | 0.807 ± 0.009 | 0.820 ±0.027* |
| 23 | 0.561 ± 0.010 | 0.601 ± 0.001 | 0.583 ± 0.028 | **0.607 ± 0.010** | 0.585 ± 0.011 |
| 28 | 0.989 ±0.006* | 0.991 ±0.000* | 0.988 ± 0.002 | 0.990 ± 0.002 | 0.987 ± 0.003 |
| 29 | 0.895 ±0.008* | 0.893 ± 0.012 | 0.895 ± 0.016 | 0.900 ±0.001* | 0.878 ± 0.003 |
| 31 | 0.756 ±0.017* | 0.767 ± 0.011 | 0.780 ±0.012* | 0.732 ± 0.015 | 0.757 ± 0.004 |
| 32 | 0.994 ±0.001* | 0.993 ± 0.005 | 0.993 ± 0.005 | 0.996 ±0.001* | 0.985 ± 0.001 |
| 37 | 0.838 ±0.014* | 0.841 ± 0.010 | 0.853 ± 0.008 | 0.859 ±0.018* | 0.853 ± 0.015 |
| 43 | 0.932 ±0.004* | 0.929 ±0.005* | 0.922 ± 0.011 | 0.919 ± 0.013 | 0.917 ± 0.018 |
| 45 | 0.952 ±0.002* | 0.935 ± 0.004 | 0.943 ± 0.021 | 0.945 ± 0.010 | 0.952 ±0.003* |
| 49 | 0.972 ±0.013* | 0.978 ±0.018* | 0.950 ± 0.034 | 0.960 ± 0.035 | 0.940 ± 0.027 |
| 53 | 0.798 ± 0.059 | **0.878 ± 0.004** | 0.869 ± 0.022 | 0.867 ± 0.016 | 0.846 ± 0.013 |
| 219 | 0.897 ± 0.035 | 0.843 ± 0.018 | **0.938 ± 0.007** | 0.856 ± 0.053 | 0.871 ± 0.049 |
| 2074 | 0.905 ±0.011* | 0.888 ± 0.008 | 0.905 ±0.012* | 0.899 ± 0.004 | 0.887 ± 0.017 |
| 2079 | 0.639 ±0.032* | 0.578 ± 0.051 | 0.648 ±0.008* | 0.631 ± 0.034 | 0.556 ± 0.023 |
| 3021 | 0.962 ±0.011* | 0.939 ± 0.017 | 0.955 ± 0.008 | 0.889 ± 0.041 | 0.958 ±0.007* |
| 3022 | 0.968 ±0.023* | 0.980 ± 0.005 | 0.982 ±0.005* | 0.966 ± 0.018 | 0.878 ± 0.085 |
| 3549 | 0.985 ±0.004* | 0.992 ± 0.000 | 0.987 ± 0.006 | 0.992 ±0.002* | 0.986 ± 0.003 |
| 3560 | 0.235 ±0.002* | 0.238 ± 0.012 | 0.213 ± 0.014 | 0.243 ±0.014* | 0.191 ± 0.007 |
| 3902 | 0.876 ±0.013* | 0.829 ± 0.028 | 0.858 ± 0.010 | 0.860 ±0.034* | 0.851 ± 0.017 |
| 3903 | 0.754 ±0.033* | 0.738 ±0.032* | 0.676 ± 0.084 | 0.737 ± 0.026 | 0.691 ± 0.033 |
| 3904 | 0.642 ±0.016* | 0.651 ± 0.017 | 0.655 ± 0.028 | 0.665 ±0.014* | 0.664 ± 0.008 |
| 3913 | 0.815 ±0.009* | 0.833 ±0.018* | 0.812 ± 0.021 | 0.790 ± 0.041 | 0.804 ± 0.019 |
| 3917 | 0.739 ±0.013* | 0.742 ± 0.020 | 0.739 ± 0.022 | 0.736 ± 0.028 | 0.754 ±0.011* |
| 3918 | 0.758 ±0.029* | 0.736 ± 0.059 | 0.761 ±0.011* | 0.732 ± 0.034 | 0.717 ± 0.033 |
| 7592 | 0.844 ±0.002* | 0.831 ± 0.004 | 0.838 ±0.015* | 0.819 ± 0.009 | 0.822 ± 0.010 |
| 9910 | 0.798 ±0.003* | 0.761 ± 0.024 | 0.774 ± 0.044 | 0.783 ± 0.017 | 0.795 ±0.003* |
| 9946 | 0.982 ±0.012* | 0.983 ± 0.005 | 0.982 ± 0.021 | 0.994 ±0.010* | 0.993 ± 0.002 |
| 9952 | 0.894 ±0.006* | 0.885 ±0.017* | 0.858 ± 0.031 | 0.831 ± 0.094 | 0.877 ± 0.019 |
| 9957 | 0.865 ±0.021* | 0.849 ± 0.003 | 0.850 ± 0.011 | 0.870 ±0.028* | 0.859 ± 0.017 |
| 9960 | 0.995 ±0.001* | 0.982 ± 0.017 | 0.942 ± 0.040 | 0.989 ± 0.007 | 0.996 ±0.001* |
| 9964 | 0.925 ±0.011* | 0.939 ±0.020* | 0.909 ± 0.043 | 0.924 ± 0.036 | 0.922 ± 0.010 |
| 9971 | 0.698 ±0.053* | 0.697 ±0.036* | 0.674 ± 0.005 | 0.668 ± 0.006 | 0.666 ± 0.034 |
| 9976 | 0.756 ±0.121* | 0.746 ± 0.131 | 0.746 ±0.125* | 0.739 ± 0.110 | 0.670 ± 0.054 |
| 9977 | 0.971 ±0.001* | 0.948 ± 0.011 | 0.967 ±0.006* | 0.936 ± 0.009 | 0.965 ± 0.002 |
| 9978 | 0.840 ±0.019* | 0.865 ±0.028* | 0.865 ± 0.005 | 0.820 ± 0.053 | 0.816 ± 0.015 |
| 9981 | 0.980 ±0.009* | 0.964 ±0.014* | 0.955 ± 0.017 | 0.953 ± 0.030 | 0.888 ± 0.003 |
| 9985 | 0.484 ±0.021* | 0.475 ± 0.013 | 0.488 ±0.011* | 0.461 ± 0.028 | 0.477 ± 0.018 |
| 10093 | 1.000 ±0.000* | 0.996 ± 0.001 | 0.993 ± 0.005 | 0.994 ± 0.007 | 0.998 ±0.003* |
| 10101 | 0.675 ±0.003* | 0.685 ±0.021* | 0.665 ± 0.049 | 0.661 ± 0.019 | 0.675 ± 0.047 |
| 14952 | 0.959 ±0.009* | 0.959 ± 0.003 | 0.956 ± 0.006 | 0.949 ± 0.022 | 0.963 ±0.002* |
| 14954 | 0.854 ±0.025* | 0.844 ±0.027* | 0.800 ± 0.040 | 0.814 ± 0.014 | 0.799 ± 0.027 |
| 14965 | 0.857 ±0.014* | 0.777 ± 0.072 | 0.856 ±0.014* | 0.838 ± 0.015 | 0.839 ± 0.006 |
| 14969 | 0.545 ± 0.028 | **0.641 ± 0.014** | 0.585 ± 0.084 | 0.546 ± 0.119 | 0.605 ± 0.004 |
| 125920 | 0.572 ±0.026* | 0.577 ±0.033* | 0.535 ± 0.024 | 0.549 ± 0.050 | 0.566 ± 0.032 |
| 125922 | 0.997 ±0.001* | 0.993 ± 0.003 | 0.996 ±0.002* | 0.993 ± 0.002 | 0.987 ± 0.017 |
| 146195 | 0.694 ±0.042* | 0.616 ± 0.111 | 0.720 ±0.029* | 0.551 ± 0.130 | 0.670 ± 0.048 |
| 146800 | 0.999 ±0.001* | 0.999 ± 0.002 | 0.979 ± 0.015 | 0.999 ±0.001* | 0.994 ± 0.008 |
| 146817 | 0.795 ± 0.021 | 0.807 ± 0.019 | **0.841 ± 0.020** | 0.808 ± 0.056 | 0.802 ± 0.036 |
| 146819 | 0.848 ±0.015* | 0.812 ± 0.039 | 0.818 ± 0.022 | 0.824 ± 0.011 | 0.836 ±0.016* |
| 146820 | 0.924 ±0.030* | 0.919 ± 0.045 | 0.963 ±0.005* | 0.875 ± 0.107 | 0.955 ± 0.008 |
| 146821 | 0.963 ±0.040* | 0.969 ± 0.021 | 0.942 ± 0.052 | 0.990 ±0.010* | 0.983 ± 0.008 |
| 146822 | 0.927 ±0.012* | 0.939 ±0.015* | 0.935 ± 0.008 | 0.916 ± 0.024 | 0.926 ± 0.003 |
| 146824 | 0.982 ±0.002* | 0.968 ± 0.012 | 0.954 ± 0.033 | 0.976 ±0.006* | 0.974 ± 0.008 |
| 146825 | 0.857 ±0.014* | 0.847 ±0.020* | 0.828 ± 0.041 | 0.839 ± 0.022 | 0.810 ± 0.056 |
| 167119 | 0.890 ±0.006* | 0.874 ± 0.036 | 0.882 ±0.026* | 0.851 ± 0.013 | 0.872 ± 0.013 |
| 167121 | **0.912 ± 0.025** | 0.726 ± 0.158 | 0.619 ± 0.003 | 0.715 ± 0.180 | 0.731 ± 0.154 |
| 167125 | 0.897 ±0.005* | 0.891 ± 0.002 | 0.889 ± 0.003 | 0.894 ±0.022* | 0.884 ± 0.004 |
| 167140 | 0.944 ±0.005* | 0.942 ± 0.006 | 0.946 ±0.004* | 0.931 ± 0.000 | 0.886 ± 0.009 |
| 167141 | 0.801 ±0.092* | 0.842 ± 0.031 | 0.838 ± 0.027 | 0.818 ± 0.027 | 0.846 ±0.027* |

Table 8: Comparative learning performances on OpenML datasets over sampling 30 configurations of the **SVM** pipeline. Performances that are statistically significant compared to the second best are in bold. Statistically comparable performances are indicated with (*). Pairwise comparisons and the associated p-value along the iterations are presented in Fig. 11.

## I   PAIRWISE COMPARISONS

Figs. 9-11 highlight a side-by-side comparison of METABU with each baseline set of meta-features. These comparisons establish the relative improvement over each baseline, that may be lost in the general comparison, Fig. 3.

On Random Forest pipeline, METABU meta-features perform on par with SCOT meta-features. Whereas its improvement over Landmark MF is only significant between the (approximately) 8th and 23rd iteration, METABU consistently outperforms Random and AutoSkLearn meta-features along the iterations.

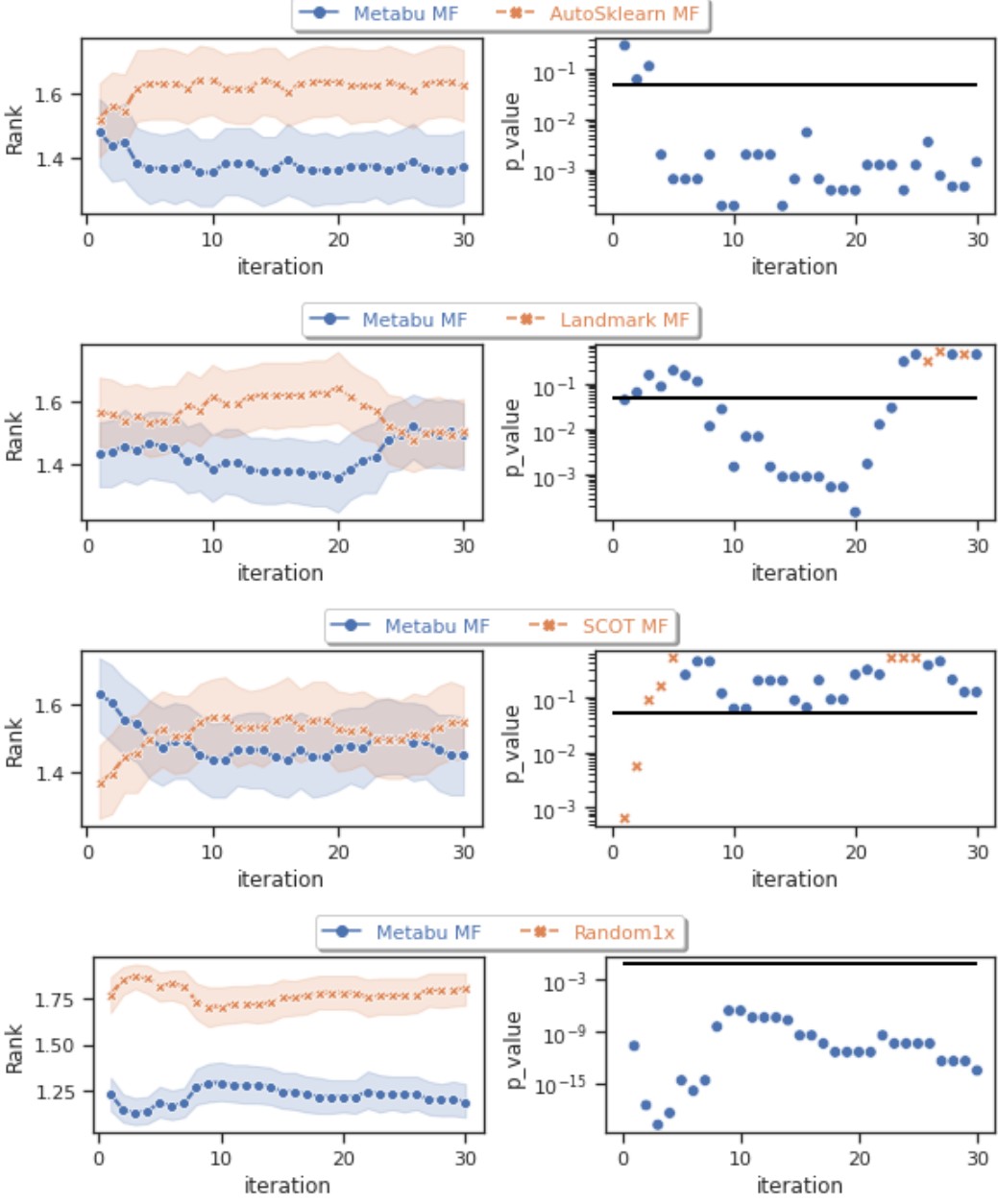

Figure 9: Pairwise comparison of METABU with baseline meta-features on **Random Forest** pipeline. Left: the average ranks. Right: the p-value assessing the statistical significance of the ranks according to the Mann-Whitney Wilcoxon test; the black horizontal line indicates the significance threshold p-value=0.05.

On Adaboost, METABU meta-features perform similarly as Landmark meta-features. Interestingly, METABU always has a better average rank than the baselines except for the first two iterations of the AutoSkLearn baseline. It is seen that the p-value is most generally below the threshold .05, establishing the statistical significance of the rank performance.

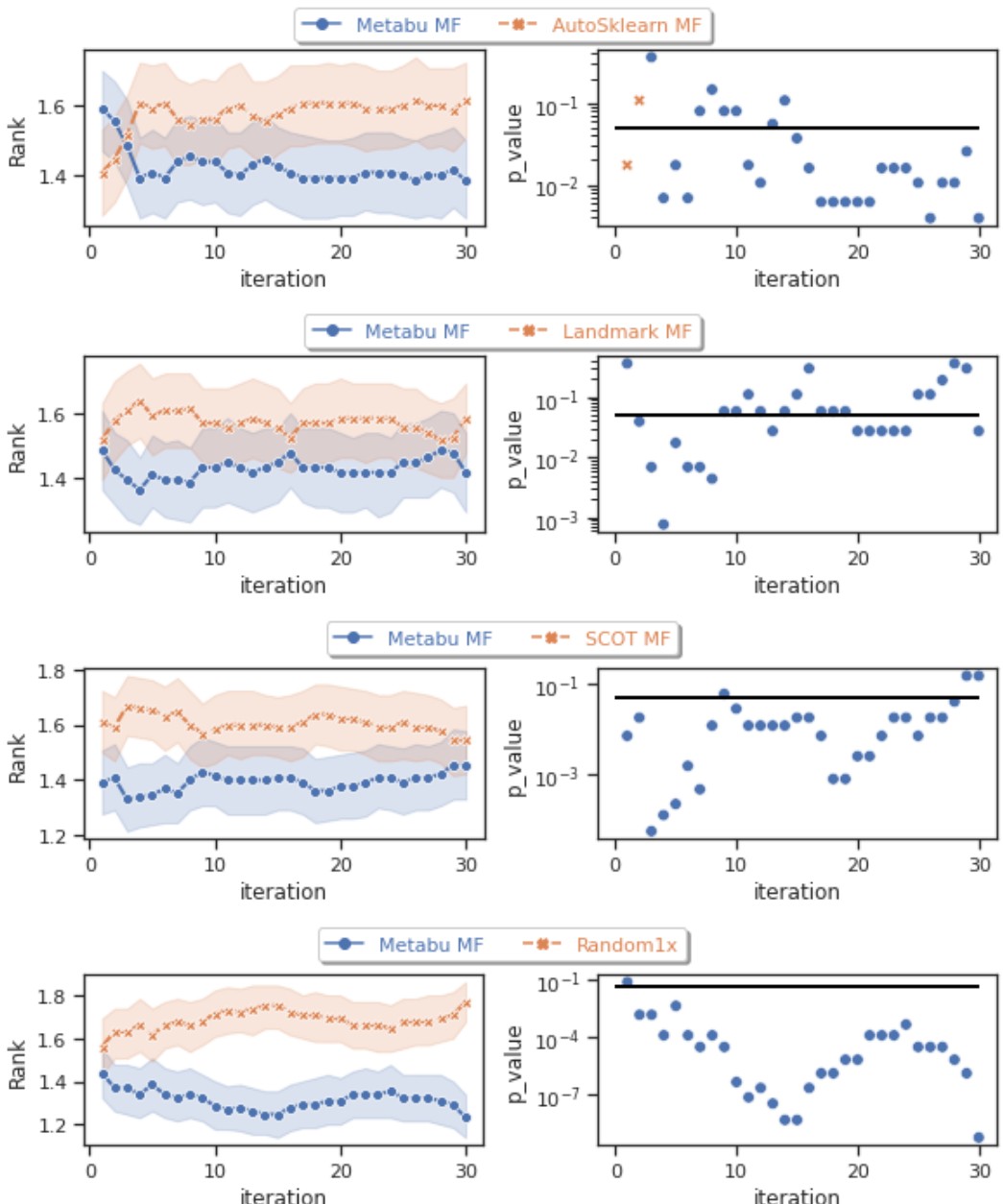

Figure 10: Pairwise comparison of METABU with baseline meta-features on **Adaboost** pipeline. Left: the average ranks. Right: the p-value assessing the statistical significance of the ranks according to the Mann-Whitney Wilcoxon test; the black horizontal line indicates the significance threshold p-value=0.05.

Lastly, the gaps in performance for SVM are striking. METABU meta-features consistently outperform all the baselines meta-features with high confidence.

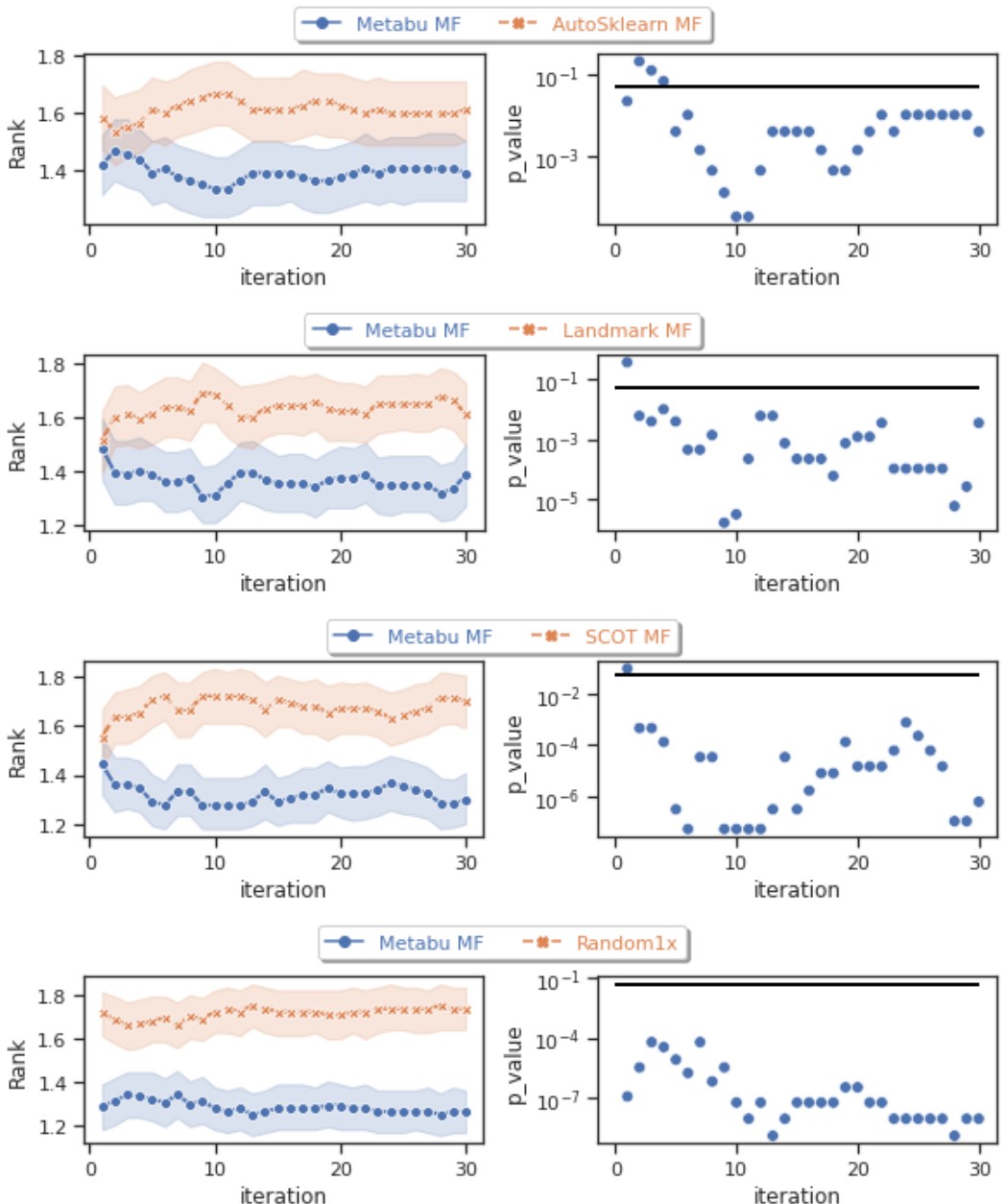

Figure 11: Pairwise comparison of METABU with baseline meta-features on **SVM** pipeline. Left: the average ranks. Right: the p-value assessing the statistical significance of the ranks according to the Mann-Whitney Wilcoxon test; the black horizontal line indicates the significance threshold p-value=0.05.

## J    SENSITIVITY ANALYSIS OF $d$

Table 9 reports the NDCG@k performance of METABU on Task 1 for varying values of $d$, showing that: i) the best results are obtained for the intrinsic dimension in the vast majority of cases; ii) the sensitivity w.r.t. $d$ is very moderate.

The intrinsic dimension $d$ of the OpenML benchmark is circa 6 for AutoSkLearn, 8 for Adaboost, 9 for RandomForest and 14 for Support Vector Machines.

| $d \setminus$ NDCG@$k$ | Random Forest | | | | Adaboost | | | | SVM | | | |
|---|---|---|---|---|---|---|---|---|---|---|---|---|
| | 10 | 15 | 20 | 25 | 10 | 15 | 20 | 25 | 10 | 15 | 20 | 25 |
| 2 | 0.57 | 0.65 | 0.71 | 0.76 | 0.6 | 0.67 | 0.73 | 0.78 | 0.55 | 0.62 | 0.68 | 0.73 |
| 5 | 0.58 | 0.65 | 0.71 | 0.75 | 0.6 | 0.67 | 0.73 | 0.78 | 0.58 | 0.65 | 0.7 | 0.74 |
| 10 | 0.57 | 0.65 | 0.71 | 0.76 | 0.63 | 0.7 | 0.75 | 0.8 | 0.58 | 0.65 | 0.71 | 0.75 |
| 15 | 0.58 | 0.66 | 0.72 | 0.76 | 0.62 | 0.7 | 0.75 | 0.79 | 0.57 | 0.65 | 0.71 | 0.76 |
| 20 | 0.58 | 0.67 | 0.73 | 0.77 | 0.62 | 0.69 | 0.74 | 0.79 | 0.58 | 0.65 | 0.71 | 0.76 |
| 25 | 0.57 | 0.65 | 0.71 | 0.76 | 0.62 | 0.69 | 0.75 | 0.79 | 0.58 | 0.65 | 0.71 | 0.76 |
| intrinsic | 0.59 | 0.67 | 0.73 | 0.78 | 0.62 | 0.69 | 0.75 | 0.8 | 0.59 | 0.67 | 0.73 | 0.78 |

Table 9: Sensitivity of METABU w.r.t the number $d$ of METABU meta-features on Task 1. The performance is the NDCG@k score measuring the relevance of the ranking induced by METABU w.r.t. the target representation.

## K  PERFORMANCE CURVES

These curves, in addition to the rank results displayed in Fig. 3b, display the performance values on 10 representative datasets from OpenML CC-18, in the context of Task 2 for respectively Random Forest (Fig. 12), Adaboost (Fig. 13), and SVM (Fig. 14). At each iteration, the curve reports the average performance value with its the standard deviation (on 3 runs).

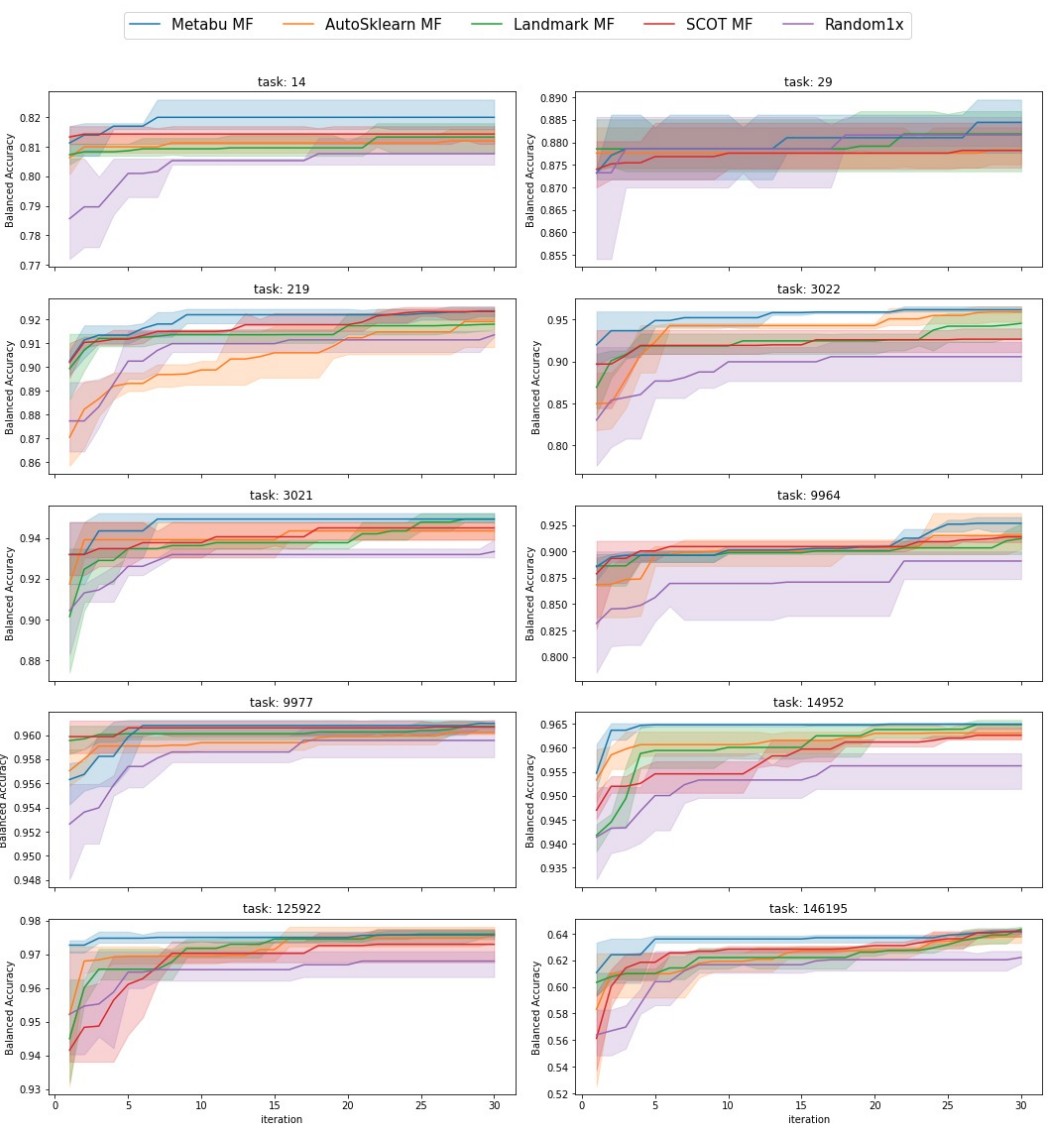

Figure 12: Performance curves on Random Forest.

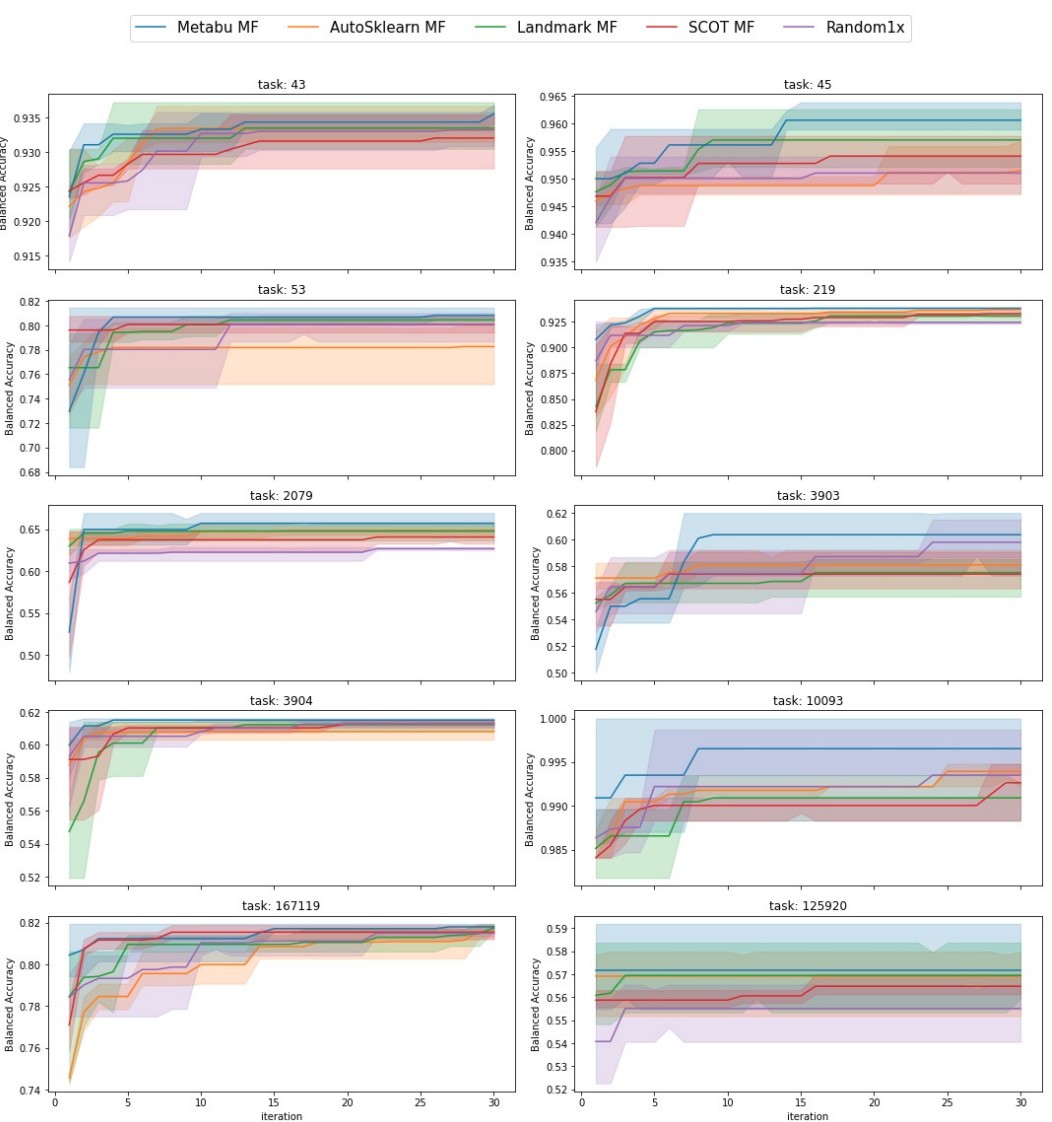

Figure 13: Performance curves on Adaboost.

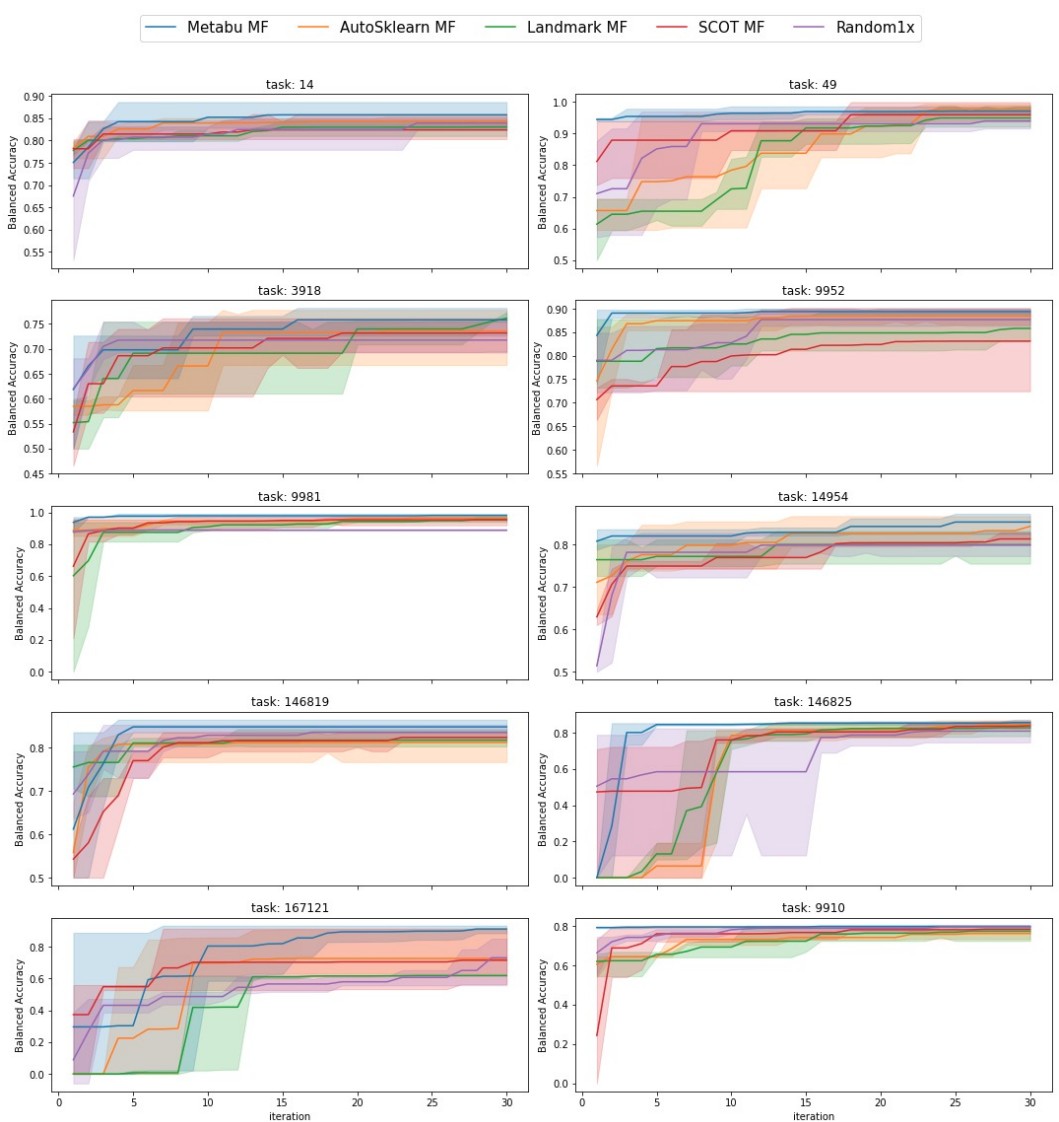

Figure 14: Performance curves on SVM.

