# OpenReview forum: "Learning meta-features for AutoML"
_ICLR.cc/2022/Conference — ICLR 2022 Spotlight_

### Official Review · Reviewer_H82u · 2021-11-02

**Correctness:** 4
**Technical Novelty And Significance:** 3
**Empirical Novelty And Significance:** 3
**Recommendation:** 8
**Confidence:** 4

**Main Review:**

# Correctness

Although I like the presented idea very much, I find the empirical evaluation far from convincing. The main reason for this is the sole usage of ranking based metrics in order to assess how much information the meta-features capture in different tasks.

Starting with task 1, it is unclear how the relevance scores, which are required to compute the NDCG are computed. This is extremely important as NDCG values will naturally vary drastically depending on how the relevance score is defined. Moreover, assessing the quality of the meta-features in terms of a ranking metric gives only limited insight on the quality of the learned embedding. While it seems that the learned meta-features are well suited for capturing proximities to datasets with similar rankings over the algorithm configurations, it is unclear how well the embedding actually captures actual performance similarities. As an illustration, consider an example with three datasets $D_1$, $D_2$ and $D_3$ and two configurations $c_1$ and $c_2$, where $c_1 (c_2)$ yields an accuracy of 0.9 (0.1) on $D_1$, 0.4 (0.1) on $D_2$ and 0.33 (0.1) on $D_3$. Clearly, $c_1$ should always be ranked before $c_2$ as it always yields the better performance. However, when considering $D_2$ as a test dataset, it very much depends on the application if one wants $D_1$ or $D_3$ to be closer to $D_2$ in the learned embedding. If one is ONLY interested in the ranking across configurations, then the choice is irrelevant and both serve the purpose. However, if one is in addition to the ranking also interested in an estimated performance (e.g. for the usage inside an acquisition function in Bayesian optimization such as in AutoSklearn), then $D_3$ should be closer to $D_2$ than $D_1$ as the performance of $c_1$ on $D_3$ is closer to its performance on $D_2$.  Depending on how the relevance score of the NDCG metric is defined, the metric might also indirectly give an insight wrt. to the aforementioned example. However, adding (in addition to the ranking metric) another metric, which quantifies performance similarities in a more direct manner, is crucial to understand the quality of the learned meta-features and should thus be added.

The main weaknesses of the evaluation can be found in tasks 2 and 3, in my opinion. Showing only rank plots in the main paper is insufficient as they hide the degree of improvement. This becomes particularly clear if one considers the actual performance tables provided in appendix H (for task 2) where one can see that on many datasets the performance differences between METABU MF and  the other methods are negligible. In fact, there seems to be almost no statistically significant differences with only a handful of exceptions. To me this is quite in contrast to the claims made by the authors in the main paper that their approach is statistically significantly better than the baselines. This might be true for the average rank, but it seems to be far from the truth when looking at the exact performances. Furthermore, there are no detailed results for task 3 at all in the appendix such that the rank plots are even less meaningful for task 3. The drastically varying p-levels in Tables 6-8 in the appendix also do not give a good impression. In my opinion, the authors should revise the evaluation to show concrete performance gains instead of rank plots. In particular, instead of showing the ranks of the methods over the time of the optimization, I suggest to plot the concrete performance of the corresponding solutions. However, from the results in the appendix I have doubts that such improvements exist in a non-negligible form.

Furthermore, it is unclear to me why (for task 3) the target distributions for AutoSklearn and PMF are different. In my eyes, the target distribution should be independent of the used AutoML method and only depend on the search space of the corresponding method. If this is meant by the authors, I strongly suggest to use the same search space for AutoSklearn and PMF as the results are hard to compare otherwise.

Lastly, I believe that some statements in the paper are formulated in a way making them at least debatable:

- p. 2: "the ML meta-features have hardly been effective to achieve AutoML (Misir & Sebag, 2017) or even to distinguish among hard and easy datasets w.r.t. a given learning algorithm (Muñoz et al., 2018)" → I believe this claim to be wrong. Several works have shown that meta-features are well suited for algorithm recommendation even in the context of ML algorithms (including AutoSklearn 2.0 [1]).
- p. 3: "By construction, it can be cheaply computed for any dataset." → Not true for landmarking features, which are notoriously expensive.
- p. 6 goal of experiments: "The first goal is to measure the performance of the METABU meta-features" → meta-features themselves cannot have a performance.
- Explanation of results
    - Task 1: "[...] the metrics based on the baselines are deterministic" → the landmarking baselines are certainly not deterministic as they also rely on training the respective ML algorithm whose performance is tracked
    - Task 2: "METABU dominates after the beginning; all approaches but the uniform sampler yield similar performances" → Contradiction. Does it dominate or is performance similar? Both statements cannot be true at the same time.
- Ethics statement: I do not believe that the goal of AutoML is to reduce the computational resources needed to get peak performance from an ML portfolio. While I agree that AutoML methods are often faster in generating a high quality ML pipeline for a given dataset than experts as some competitions have shown, I highly doubt that they use less computational resources than the corresponding human experts would.

With all of the above in mind, I believe that one cannot assess the empirical quality of the proposed methodology based on the current state of the evaluation. Accordingly, claims made regarding the quality of the methodology are not well supported, in my opinion.

## References

[1] Feurer, Matthias, et al. "Auto-sklearn 2.0: The next generation." arXiv preprint arXiv:2007.04074 (2020).

# Novelty and Significance of Technical Algorithms, Models, or Theory

While the proposed methodology certainly is novel in the specific context of AutoML, it is well known in other contexts as acknowledged by the authors. I believe this to be completely okay for an empirical field such as AutoML. However, I believe that the authors could have focused more on giving explanations on other embedding approaches from the context of AutoML. Although the authors point to PMF and OBOE, the differences of the embedding learned in this work compared to the other embedding approaches should be explored more thoroughly on a methodological level. Thus, while the work is well contextualized wrt. to existing literature on a broad level, it lacks a good contextualization on a more nuanced level.

The significance of the methodology is hard to assess as the empirical evaluation is not reliable as stated under correctness. However, assuming that the methodology does indeed yield good meta-features, I deem it significant for the AutoML community.


# Novelty and Significance of Empirical Advancements, Insights, or Datasets

Since I have doubts regarding the empirical evaluation as a whole in this work, I deem it to be non-significant. However, the results presented in 4.4 are certainly interesting and I have not seen an analysis of the intrinsic dimension of the OpenML CC-18 benchmark before. Thus, at least this part of the evaluation yields novel insights.


# Minor Issues

- different writings of AutoSklearn throughout the paper
- in contrast with → in contrast to
- "OT, the Wasserstein distance [...]" → OT seems to be out of place there.
- p. 4: difference in notation between step 1 and step 2 - in step 1 "d" is the 2-Wasserstein distance whereas in step 2 "d_W^2" is the distance.

-----
# Update after Rebuttal

I would like to thank the authors for their strong rebuttal and excuse myself for misjudging that the learned embedding is performance preserving instead of rank preserving rendering my criticism of the evaluation task 1 infeasible. My remaining questions have been thoroughly answered and the requested changes to the evaluation have been added to the paper where applicable. As my main concern has been with the evaluation, all reasons why I could not suggest acceptance of this paper have been ruled out. Although I do not agree to all opinions presented in the paper and the rebuttal (e.g. computational resources), I do not have any further factual criticism.  As such, I will adjust my rating correspondingly to suggest acceptance of this work.

**Summary Of The Paper:**

The paper presents an approach, called MetaBu, for learning a meta-feature embedding from an existing meta-feature space into a latent space, which is aims at being rank preserving regarding different hyper-parameter configurations. The special kind of embedding and its property of aiming at being performance preserving in the context of AutoML is the main contribution of the paper, in my opinion. The quality of the learned meta-features is assessed through different experiments such as capturing to what degree the embedding is indeed performance preserving and how well AutoML tools perform when initialized with the corresponding meta-features. Moreover, the authors provide a sensitivity analysis of relevant hyper-parameters of MetaBu and demonstrate how to gain insights from the learned embeddings.

**Summary Of The Review:**

Overall I like the presented idea very much and I deem the paper to be overall well written, except for some debatable statements (see Correctness). As such, both the methodology and the experiments are well described. Unfortunately, the part of the evaluation assessing the quality of the learned meta-features is insufficient (see Correctness) and needs to be revised in my opinion. Accordingly, I cannot recommend acceptance of this work in its current form.

-----
# Update after Rebuttal
All my concerns have been fixed or ruled out by the authors in the revised version. I recommend to accept the paper in its revised form.

---

> ### Author Response · Authors · 2021-11-16
> **Comparing performance ranks vs performance values.**
>
> Q: The paper presents an approach, called MetaBu, for learning a meta-feature embedding from an existing meta-feature space into a latent space which is aims at being performance preserving regarding different hyper-parameter configurations
>
> A: We are afraid that this summary does not reflect the paper:  the embedding aims to be rank-preserving w.r.t. the hyper-parameter configurations (as opposed to performance-preserving).
>
> -----
>
> Q: Starting with task 1, it is unclear how the relevance scores, which are required to compute the NDCG, are computed.
>
> A: We use the mainstream definition to assess information retrieval approaches:
> $$DCG(\mbox{dataset }E, k) = \sum_i^k \frac{2^{rel (E_i, k)} -1}{log_2 (i+1)}$$
> with $E_i$ the $i$-th nearest dataset to $E$ according to the target representation, and $rel(E_i, k)$ is a binary function returning 1 if $E_i$ is ranked in the top-$k$ according to the Euclidean metric on the assessed meta-features (AutoSkLearn, Landmark, SCOT or Metabu meta-features) else 0.
> The reported NDCG@k normalizes the above $DCG(E, k)$ and averages the normalized DCG over all datasets $E$.
>
>
> -----
>
> Q: While it seems that the learned meta-features are well suited for capturing proximities to datasets with similar rankings over the algorithm configurations, it is unclear how well the embedding actually captures actual performance similarities.
>
> A You are right, our goal is to capture the similarities among top-ranked configurations, as opposed to the similarities among the performances. Actually the state of the art in AutoML for over 10 years (e.g. [Bergstra11,Bardenet13,Guyon19,Liu21], see below)
> leverages performance ranks as opposed to performance values for the following reasons:
> 1. performances are noisy due to the sampling of the training and validation sets, and possibly the algorithm noise;
> 2. AutoML is an optimization problem in the sense that it only aims to capture the top configurations; it is not a modeling problem (misleading a very bad configuration for a bad one makes no difference).
>
>
> -----
>
> Q: ... Showing only rank plots in the main paper is insufficient as they hide the degree of improvement.
>
> A: We respectfully disagree with the reviewer's point of view. Numerous AutoML papers use rank plots in their evaluations  (e.g. [Bardenet13,Feurer15,Feurer19,Rakotoarison19,Yang19,Guyon19,Liu21], see below). An interesting aspect of ranks is that they are dependent on the performance of every competitor, accounting for the competitiveness w.r.t. the other algorithms and the difficulty of the datasets.
>
> To better assess the degree of improvement, complementary results comparing Metabu and each baseline side-by-side are reported in Appendix-I, Figures 9-11.
>
> -----
>
> Q: To me this is quite in contrast to the claims made by the authors in the main paper that their approach is statistically significantly better than the baselines.
>
> A: As the results in Tables 6-8 only display performances at the 30th iteration, we added the results along iterations:
> The added Figures 9-11 (Appendix-I) report the side-by-side comparison of Metabu with each baseline, with the p-value of this rank-based comparison (using the Mann-Whitney Wilcoxon test) for iterations 1 to 30.
> These figures show that Metabu outperforms each competitor (lower rank) and the p-values are almost always below 0.05, except for baseline SCOT in the RandomForest case.
>
>
>
>
> -----
>
> Q: I suggest to plot the concrete performance of the corresponding solutions.
>
> A: Thanks for this suggestion. We added Figures 12-14 in Appendix K, reporting the performance plots for 10 datasets in the RandomForest, Adaboost and SVM cases.
>
>
> -----
>
> Q: it is unclear to me why (for task 3) the target distributions for AutoSklearn and PMF are different.
>
> A: You are right, the target representation does not depend on the AutoML method.
>
> However the search space of the 2019 version of AutoSklearn is not exactly the same as the one used in PMF (2017). We thus decided to report the results along two different plots.
>
> Also, the point of Task 3 is to highlight the improvement of AutoSklearn and PMF when replacing their meta-features with Metabu meta-features.
>
>
> -----
>
> Q: p. 2: "the ML meta-features have hardly been effective to achieve AutoML (Misir & Sebag, 2017) ..." → I believe this claim to be wrong.
>
> A: We shall mitigate this claim in the revised version. What we intended to say was: if we had very effective meta-features, then a basic nearest-neighbor approach would do the job and deliver a good AutoML solution (retain the same top configurations as for the dataset nearest to the current dataset).
>
> But the lesson learned from e.g., AutoSkLearn is that we need to consider other configurations than the ones based on this nearest neighbor approach; these additional configurations are provided by Bayesian Optimization.

---

> > ### Author Response · Authors · 2021-11-16
> > **Answer to Reviewer H82u  (next)**
> >
> > -----
> >
> > Q: p. 3: "By construction, it can be cheaply computed for any dataset." → Not true for landmarking features ...
> >
> > A: In our experiments, we only used cheap landmarking features, such as based on decision tree, LDA, and naive Bayes. An overview of the computational time is displayed in Fig 7, Appendix F.
> >
> >
> > -----
> >
> > Q: "The first goal is to measure the performance of the METABU meta-features" → meta-features themselves cannot have a performance.
> >
> > A: Thank you. We clarified it: "The first goal is to assess the dataset neighborhood induced by the METABU meta-features".
> >
> >
> > -----
> >
> >
> > Q: Task 1: "[...] the metrics based on the baselines are deterministic" → the landmarking baselines are certainly not deterministic ...
> >
> > A: You are right. However in the experiments, the considered landmarking features involve deterministic algorithms (decision trees, LDA and Naive Bayes).
> >
> > -----
> >
> > Q: METABU dominates after the beginning; all approaches but the uniform sampler yield similar performances" → Contradiction.
> >
> > A: You are right, thank you for pointing this out - as a matter of fact, one word was missing:
> > METABU dominates after the beginning; all *other* approaches but the uniform sampler yield similar performances.
> >
> > -----
> >
> > Q: I highly doubt that they use less computational resources than the corresponding human experts would.
> >
> > A: May we respectfully disagree? The AutoML trend is gaining momentum for over 10 years, and it is acknowledged as a priority for ML to reach out to applications. This suggests that either there is a shortage of human experts, or that human experts might stop exploring the configuration space before reaching peak performance. Overall, human experts might use less resources but the question of arriving at the peak performance remains.
> >
> >
> > -----
> >
> > Q: I believe that the authors could have focused more on giving explanations on other embedding approaches from the context of AutoML.
> >
> > A: This is clarified (p. 2):
> > Metabu learns once for all new meta-features (embedding), aimed to capture the topology (neighborhoods) corresponding to the target representation. These meta-features are computable from scratch for each new dataset. In contrast,  Yang et al. (2019) and Fusi et al. (2018) both require a cold-start phase,
> > launching configurations for a new dataset and using their performance to find the representation of the new dataset.
> >
> > The Metabu meta-features can be used to select configurations (as in Task 2); they can also be used to initialize the PMF / OBOE search (as in Task 3).
> >
> >
> > -----
> >
> > [Bergstra11] James Bergstra, R. Bardenet, Yoshua Bengio, and Balázs Kégl. Algorithms for hyper-parameter optimization. NIPS 2011, volume 24
> >
> >
> > [Bardenet13] Rémi Bardenet, Mátyás Brendel, Balázs Kégl, and Michèle Sebag. Collaborative hyperparameter tuning. ICML 2013, pp. 199-207.
> >
> > [Feurer15] Matthias Feurer, Aaron Klein, Katharina Eggensperger, Jost Springenberg, Manuel Blum, and Frank Hutter. Efficient and robust automated machine learning. NIPS 2015, pp. 2962–2970.
> >
> >
> > [Hutter19] Frank Hutter, Lars Kotthoff, and Joaquin Vanschoren (eds.). Automated Machine Learning: Methods, Systems, Challenges. The Springer Series on Challenges in Machine Learning. Springer, 2019.
> >
> > [Guyon19] Isabelle Guyon et al. Analysis of the AutoML challenge series 2015-2018. In Automated Machine Learning: Methods, Systems, Challenges, Springer 2019.
> >
> > [Yang19] Chengrun Yang, Yuji Akimoto, Dae Won Kim, and Madeleine Udell. OBOE: collaborative filtering for Automl model selection. In ACM SIGKDD International Conference on Knowledge Discovery & Data Mining, pp. 1173–1183. ACM, 2019
> >
> > [Rakotoarison19] Herilalaina Rakotoarison, Marc Schoenauer, Michèle Sebag: Automated Machine Learning with Monte-Carlo Tree Search. IJCAI 2019: 3296-3303
> >
> > [Feurer19] Matthias Feurer, Aaron Klein, Katharina Eggensperger, Jost Tobias Springenberg, Manuel Blum, Frank Hutter: Auto-sklearn: Efficient and Robust Automated Machine Learning. Automated Machine Learning 2019: 113-134
> >
> > [Liu21] Zhengying Liu et al., Winning Solutions and Post-Challenge Analyses of the ChaLearn AutoDL Challenge 2019. IEEE Trans. Pattern Anal. Mach. Intell. 43(9): 3108-3125 (2021)

---

### Official Review · Reviewer_A4JL · 2021-11-02

**Correctness:** 3
**Technical Novelty And Significance:** 3
**Empirical Novelty And Significance:** 3
**Recommendation:** 6
**Confidence:** 3

**Main Review:**

Overall, I think the main idea of the paper is quite intriguing. However, due to issues with the presentation, I had a hard time to follow the paper and understand what is actually done. Still, I am not fully convinced I was really able to catch the paper in its entirety.

Furthermore, I could identify some statements or claims, which are either wrong or not adequately supported (either in terms of a reference or some theoretical/empirical proof):
- AutoML is the problem of automating the entire data engineering process, not only the model selection and HPO stage.
- "Neural networks notoriously need large amounts of samples to be efficiently trained, [...]": I am pretty sure that the efficiency of the training does not directly depend on the number of samples.
- "By construction, it can be cheaply computed for any dataset." => Meta features are not always cheap to compute. Especially, landmarking meta-features can become quite expensive.
- OpenML CC-18 is referred to as the largest curated tabular dataset benchmark. However, there is a dataset published with the publication of Fusi et al. providing performance data across roughly 600 data sets.
- "the best regions in hyper-parameter space are the same for most datasets" => The authors do not provide any support for this claim.


Detailed coments:
- The authors refer to two sets of meta features and call them "the basic one" and "the target one". First of all, it takes some time to notice that these are actually the names. Secondly, the explanation of "the target one" is not understandable to me.  How does such a distribution of the hyper-parameter configurations of A look like?
- "OpenML CC" => OpenML CC-18
- what does it mean "to achieve AutoML in the context of AutoSklearn"?
- The definition of Optimal Transport introduces a variable q which is first not used but then suddenly appears in the expected value.
- "a one-hote encoding" => an one-hot encoding
- the explanation of what "\psi" does is not understandable. The sentence starts with "In brief, mapping \psi sends the naive meta-feature[...]"
- How is the bootstrapping done that 1,0000 x n training data sets are given to the learning algorithm?
- What is "the AutoML selection problem"?
- What is meant by "learning performance data"?
- What is considered to be "enough learning performance data"? There is no explanation on that.
- The significance test for the performances first of all assumes a normal distribution and further that the samples are independent (which they are not, since the samples are paired with respect to the test data set). Therefore, this test cannot be applied here.

# Update after Rebutttal

After reading the other reviews, the rebuttal, and the updated version of the paper, my concerns regarding the paper vanished. Furthermore, considering the recommendations of the other reviewers it seems that I am the outlier here, so I updated my score accordingly. Still I would like to note that AutoML is not only about algorithm selection and hyper-parameter optimization. The model selection stage is what current AutoML systems are capable of but this is only a small part of the whole AutoML vision which aims to automate the entire data science process and not only model selection.

**Summary Of The Paper:**

The paper "Learning meta-features for AutoML" proposes an approach to learn a linear combination of data set meta features. To this end, the problem of learning good meta-features is tackled via an optimal transport problem considering the data set meta features as well as the top performing hyper-parameter configurations of a learning algorithm.



**Summary Of The Review:**

While the results seem to be very promising, due to the issues raised above, I feel that this paper is not ready for publication yet. Therefore, I cautiosly recommend reject.

# Update after Rebuttal
Due to the strong responses by the authors as well as the updated version of the paper I recommend accepting the paper.

---

> ### Author Response · Authors · 2021-11-16
> **(Meta-)learning requires (sufficiently many) meta-examples. The number and quality of datasets in OpenML and OpenML CC-18. Significance test.**
>
> Q1: AutoML is the problem of automating the entire data engineering process ...
>
> A: You are right, AutoML is a data engineering pipeline, including e.g. the data preparation stage before the model selection and HPO stage.
> However, as done in AutoSkLearn, automating this pipeline process can be formalized as a single Combined Algorithm Section and Hyper-parameter optimization (CASH) problem: the choice of algorithm at every stage is achieved by setting a hyper-parameter.
> Importantly, we believe that our work covers the full AutoML process: the considered search space in AutoSKlearn (Feurer et al. 2015) and PMF (Fusi et al. 2017) includes the stages of data cleaning, data reduction, feature selection, besides the learning algorithms.
>
> -----
>
> Q2: I am pretty sure that the efficiency of the training does not directly depend on the number of samples.
>
> A: Indeed there are many factors governing the efficiency of the training.  However, the Data Augmentation issue has been gaining increasing attention recently in Machine Learning, suggesting that huge performance gains can be obtained by considering more examples.
>
>
> -----
>
> Q: Meta features are not always cheap to compute.
>
> A: The most expensive meta-features are based on landmarking. However we used cheap models such as decision trees, naive Bayes, LDA, with a negligible cost compared to the overall computational cost of the hyper-parameter optimization part. Fig 7 (Appendix F) provides a comparison of the computational time of Metabu (for extracting basic meta-features, learning Metabu meta-features) with the training of *one* classifier.
>
> -----
>
> Q: Fusi et al. providing performance data across roughly 600 data sets
>
> A: At the time of writing,  OpenML contains 3,448 datasets to our best knowledge. However, some of them have data quality issues (with e.g., datasets with some constant features; some with a single feature; some datasets are too big or ill-conditioned, entailing a large SVM running time) and some datasets are deprecated versions of the others, which may create a risk of over-optimistic leave-one-out evaluation.
> Because of this, Bischl et al. proposed OpenML CC-18, a curated benchmarking suite for AutoML. As far as we know, OpenML CC-18 is the largest *curated* tabular dataset benchmark available for AutoML.
>
>
> -----
>
> Q: "the best regions in hyper-parameter ..." => The authors do not provide any support for this claim
>
> A: Actually, this paragraph was not intended as a claim: it contrasts the two extreme cases (very diverse best configurations vs same best configuration for the different datasets) for an insight into the intrinsic dimension of the benchmark (higher for the former than for the latter case). The ambiguity is removed in the revised version.
>
>
> -----
>
> Q: How does such a distribution of the hyper-parameter configurations of dataset $A$ look like?
>
> A: The different representations are illustrated in Fig 2.
>
> ** The target representation ${\bf z}\_A$ of dataset $A$ refers to the discrete distribution of the $L$-top configurations for $A$, denoted $\Theta_A$. Formally, ${\bf z}\_A=\frac{1}{L}\sum_{\theta\in\Theta_A} \delta_{\theta}$, with $\delta_{\theta}$ be the Dirac function at $\theta$.
>
> ** The basic (or naive) representation of a dataset is the vector of its hand-crafted meta-features.
>
> ** The Metabu representation of a dataset is the vector of the learned meta-features (linear combinations of the basic meta-features) for this dataset.
>
> -----
>
> Q: what does it mean "to achieve AutoML in the context of AutoSklearn"?
>
> A: We meant: achieve AutoML (solve the Combined Algorithm Section and Hyper-parameter optimization) in the context of the AutoSkLearn space,  which applies HP optimization onto the full ML pipeline (from data processing to ML algorithm). This is made clearer in the revised version.
>
> -----
>
> Q: The definition of Optimal Transport introduces a variable q ...
>
> A: Thank you. We moved the definition of $q$ near its usage.
>
> -----
>
> Q: the explanation of what "\psi" does is not understandable.
>
> A: $\psi$ is a linear mapping defined on the basic meta-features. Formally the Metabu meta-features are linear combinations of the basic meta-features, defined according to $\psi$. $\psi$ is trained such that the Euclidean distance between two datasets according to the Metabu meta-features is close to the distance between the target representation of both datasets (optimal transport).
>
> -----

---

> > ### Author Response · Authors · 2021-11-16
> > **Answer to Reviewer A4JL (next)**
> >
> > Q: "enough learning performance data"?
> >
> > A: Meta-learning exploits a meta-dataset made of triplets (dataset description, ML configuration, learning performance), referred to as meta-samples, gathered from OpenML CC-18.
> >
> > "Enough learning performance data" refers to the fact that, as discussed in Q.2, (meta-)learning needs sufficiently many (meta-)samples.
> > Further, the informative meta-samples are those with high learning performance; those with bad learning performance are not helpful to characterize the target representation of  datasets.
> > This is the reason why we used bootstrapping.
> >
> >
> > -----
> >
> > Q: How is the bootstrapping done that 1,0000 x n training data sets are given to the learning algorithm ?
> >
> > A: Bootstrapped datasets generated from a given dataset $E$ (as detailed in footnote 4) have different basic representations, and their target representation is set to the target representation of the original dataset $E$. They can thus be used to augment the meta-dataset. This bootstrapping process is meant to achieve (meta-)data augmentation, by injecting noise over the basic representation (to avoid over-fitting $\psi$). The point is that this noise is relevant to meta-learning: instead of creating an augmented dataset by perturbing the representation of an initial dataset with Gaussian noise, we create an augmented dataset by bootstrapping an initial dataset and we compute from scratch its basic representation.
> >
> >
> > -----
> >
> > Q: What is "the AutoML selection problem"?
> >
> > A: We meant: the AutoML problem. Thank you for pointing this out, we modified the paper accordingly.
> >
> > -----
> >
> > Q: The significance test for the performances ...
> >
> > A: Thank you for pointing this out. The aggregation of the performances of different configurations for the same dataset (continuous values, not independent) is now compared using Wilcoxon rank-sum. The new results show that this does not modify the respective performance of the competitors.

---

### Official Review · Reviewer_f2By · 2021-11-02

**Correctness:** 3
**Technical Novelty And Significance:** 4
**Empirical Novelty And Significance:** 3
**Recommendation:** 8
**Confidence:** 4

**Main Review:**

Strengths:

- This is a well written paper for the most part, proposing a very interesting meta-learning technique. It is relatively easy to follow from my perspective.
- The proposed method combines two forms of meta-learning: one that solely leverages the dataset meta-features to develop a notion of similarity between datasets, and use that information to seed the AutoML; and the other that does not focus on the dataset meta-features  but solely relies on the performance of similar hyperparameters on different datasets to implicitly define similarities between pairs of datasets.
- This method focuses on the general AutoML problem for tabular data where we cannot leverage well-established techniques used for AutoML in deep learning.
- The proposed scheme utilizes a linear transformation of existing interpretable meta-features, allowing for interpretation of dataset landscapes and differences between the AutoML problems with different ML models.



Weaknesses:

- While most of the paper was quite easy to follow, the short paragraphs on the "intrinsic dimension of the space of datasets" and the corresponding empirical evaluation and discussion (in section 4.4) is a little hard for me to follow. I realize that partly the issue is the novelty of the considered problem -- usually we are thinking about the intrinsic dimension of the dataset, but here we need to focus on the intrinsic dimension of a meta-dataset of datasets with respect to a particular AutoML problem, where a dataset is a "sample" in this meta-dataset, and the intrinsic dimension of this meta-dataset somewhat quantifies the difficulty of the AutoML problem for a particular AutoML problem (that is, a hyperparameter optimization for a specific ML model). However, the empirical results and discussion seem counterintuitive to me -- to me the AutoML problem in AutoSklearn (which is solving a Combined Algorithm Selection and Hyperparameter optimization or CASH problem) is a harder AutoML problem than the AutoML problem for SVM. In fact, the AutoML problem of SVM is part of the AutoML problem in Auto-sklearn. The search space of Auto-sklearn contains the search space of SVM (and RandomForest and Adaboost). So such a result confuses my interpretation of the learned meta-features and the meta-dataset of datasets. I thought the most "flexible" AutoML problem will also be the hardest, not the easiest.


**Summary Of The Paper:**

This paper focuses on the AutoML problem for tabular data and proposes a meta-learning based novel solution. They consider the optimal transport to define distances between two datasets, utilizing the Wasserstein-Gromov distance between the distribution of the top performing hyperparameters for the respective datasets. Given this distance, they propose learning a linear transformation of existing dataset meta-features such that the Euclidean distance between a pair of datasets in this transformed space is proportional to their Wasserstein-Gromov distance. This method is termed Metabu.

The empirical evaluation compares Metabu to existing meta-learning schemes on (i) their ability to capture the desired Wasserstein-Gromov distance, (ii) their ability to find better hyperparameters via sampling without an underlying optimizer, and (iii) their ability to find better seed hyperparameters for hyperparameter optimizers. The results on the OpenML CC-18 suite with 3 machine learning models indicate that Metabu significantly improves upon existing meta-learning schemes. The paper also demonstrates how the learned linear transformation of existing dataset meta-features allow us understand the importance of different existing dataset meta-features and how these vary between machine learning models.


**Summary Of The Review:**

Overall, I would like to recommend this paper for an accept for the following reasons:

- I think this is a well-written paper that positions itself well relative to existing work, and presents the novel scheme is a clear and easy-to-follow manner.
- The paper presents a significantly new meta-learning scheme building upon existing meta-learning techniques. Hence the novelty is significant in my opinion.
- The empirical evaluation is done against valid baselines, and the proposed scheme shows significant margins of improvement.

---

> ### Author Response · Authors · 2021-11-16
> **About the intrinsic dimension of a set of datasets w.r.t. an ML algorithm.**
>
> Thank you for your most insightful review.
>
> Q: to me the AutoML problem in AutoSklearn [...] is a harder AutoML problem than the AutoML problem for SVM [as the] AutoML problem of SVM is part of the AutoML problem in AutoSklearn. The search space of AutoSklearn contains the search space of SVM (and RandomForest and Adaboost).
>
> So [the fact the the intrinsic dimension is higher for SVM than for AutoSkLearn] confuses my interpretation of the learned meta-features and the meta-dataset of datasets. I thought the most "flexible" AutoML problem will also be the hardest, not the easiest.
>
> A: Thank you for raising this issue. Our intuition was the same as yours; it was puzzling to see that the intrinsic dimension is higher for SVM than for AutoSkLearn, while, as you said, solving the AutoML problem for SVM is part of solving the AutoML problem for AutoSkLearn.
>
> Our tentative interpretation is the following:
> 1. AutoSklearn does not need to fully solve the AutoML problem for SVM. If it could determine that SVM is definitely not the best algorithm in some region of the dataset representation, it could safely ignore SVM.
>
> 2. It is true indeed that the AutoSklearn search space includes that of SVM. However the difficulty of an optimization problem is not always directly related to the dimension of its search space. In some cases, as illustrated e.g., in SVM with the so-called kernel trick, or in deep neural networks known to be over-parameterized, increasing the dimension of the search space might decrease the fraction of local optima and thus facilitate the optimization (via gradient search).

---

> > ### Comment · Reviewer_f2By · 2021-11-19
> > **Thank you for response; does not help with interpretation**
> >
> > I want to thank the authors for their response.
> >
> > Given the tentative explanation, I still have a hard time with interpreting the "intrinsic dimension". If the intrinsic dimension is an indication of the hardness of the AutoML problem, the points 1 and 2 do not still seem intuitive to me. Neither of them tell me why the AutoSklearn problem is "easier" than the SVM-HPO problem.
> >
> > If the intrinsic dimension is an indication of how a particular solver is traversing the search space (regardless of the search space size), that could be intuitive: AutoSklearn seeds the search with meta-learning and portfolio building, which heavily favour gradient boosted trees or random forests (especially for relative small optimization budgets) -- so even if its search space is larger, it is always gravitating to gradient boosted trees with a small portfolio of hyperparameters, then this implicit pruning of the search space implies that the actual HPO problem AutoSklearn ends up solving is very small.

---

> > > ### Author Response · Authors · 2021-11-24
> > > **Insight on intrinsic dimension.**
> > >
> > > Thank you for raising the point of the dynamics of the solver.
> > >
> > > We propose to distinguish two sources of difficulty for AutoML:
> > > * the difficulty of the optimization / search task (using e.g. SMAC);
> > > * the difficulty related to the initialization of the search (as in AutoSkLearn or PMF), using meta-learning and the neighborhoods based on the meta-features. As you said, if the initialization is such that the search eventually explores a small part of the search space, this makes the optimization problem much easier.
> > >
> > > In AutoSklearn, as you said, some pipelines are generally good (winners). This tends to make the target representation of datasets less diverse, and equivalently, this decreases the intrinsic dimensionality.
> > > The fact that there exist such winners also make the initialization task easier, everything else being equal.
> > > This reasoning suggests that a lower intrinsic dimensionality is correlated with an easier initialization task.
> > >
> > > The above argumentation does not say anything about the difficulty of the optimization task standalone (not considering the initialization). In the case of a more complex search space, reflecting the higher flexibility of the system (as for AutoSkLearn), we believe at the moment that the optimization task standalone is more difficult, independently of the intrinsic dimensionality of the target representation.
> > > Lesion studies will be conducted to investigate this further.

---

### Official Review · Reviewer_paJ4 · 2021-11-03

**Correctness:** 3
**Technical Novelty And Significance:** 3
**Empirical Novelty And Significance:** 3
**Recommendation:** 6
**Confidence:** 3

**Main Review:**

Strengths:
- The paper is well-written and well-organized.
- The proposed method has some novelty and will contribute to the AutoML community. Learning meta-features for improving AutoML searching space is an interesting direction.
- Experimental results prove the effectiveness of learned meta-features.

Weakness:
- Page 4-5 mentions $d$ is a main hyper-parameter of METABU and is identified using an intrinsic dimensionality procedure. But the actual values of $d$ (estimated using Facco et al., 2017) in the experiments are not reported. Since this is listed as a contribution, in the experiments, it is better to assess the sensitivity of METABU w.r.t. $d$.
- The method has been only tested on traditional classifiers. It would be better to add experiments to neural networks.
- The proposed method assumes the relationship between target space and meta-features is linear. Does it make sense? Is it possible to use Kernel methods or non-linear MLP to replace OT?
- What does "topology" mean in the paper? It is not well-defined in the paper.

**Summary Of The Paper:**

This paper tackles the AutoML problem. It proposes to learn a linear combination of manually designed meta-features, which aligns meta-features with the space of hyper-parameter configurations via an Optimal Transport procedure. Experiments on OpenML benchmark demonstrate the power of the proposed method on boosting AutoML systems.

**Summary Of The Review:**

The topic and claims in this paper may be interesting to AutoML community. It contains novelty and was well written.
Overall, this is a good paper, which is marginally above the acceptance threshold.

---

> ### Author Response · Authors · 2021-11-16
> **Sensitivity analysis, Metabu components.**
>
> Q: d is a main hyper-parameter of METABU and is identified using an intrinsic dimensionality procedure. But the actual values of  (estimated using Facco et al., 2017) in the experiments are not reported. Since this is listed as a contribution, in the experiments, it is better to assess the sensitivity of METABU.
>
> A: The intrinsic dimension associated with the considered ML algorithms and pipelines are reported and discussed in Section 4.4. The intrinsic dimension is 6 for AutoSKlearn, 8 for Adaboost, 9 for Random Forest, and 14 for SVM.
>
> We provided an additional sensitivity analysis of Metabu performance w.r.t $d$ in Table 9.
>
>
> -----
> Q: The method has been only tested on traditional classifiers. It would be better to add experiments to neural networks.
>
> A: This work focuses on tabular datasets, which explains our choice to focus on traditional classifiers. Meta-learning for deep learning models is an area of research with specific issues (ranging from few-shot learning to domain adaptation), often targeting images, videos, text, or audio datasets.
>
> Nevertheless, AutoSklearn includes a (shallow) neural net in its algorithm portfolio, associated with the usual hyper-parameters (number of neurons, number of layers, type of activation function, regularization parameter, learning rate, and number of iterations).
>
> -----
> Q: The proposed method assumes the relationship between target space and meta-features is linear. Does it make sense? Is it possible to use Kernel methods or non-linear MLP to replace OT?
>
> A: The restriction to linear mappings mostly aims to avoid over-fitting. Early experiments show that i) non-linear embeddings hugely over-fit the (meta-) training set due to the small size of the (meta-) dataset; ii) Siamese networks are hard to calibrate (especially for the negative sampling) whereas Optimal Transport efficiently accounts for the close neighborhood of each dataset.
>
>
> -----
> Q: What does "topology" mean in the paper? It is not well-defined in the paper.
>
> A: "Topology" refers to the neighborhood of each dataset in the benchmark.
> The core of MetaBu is to learn meta-features and to exploit the Euclidean metric based on these meta-features. The word "topology" is used instead of "metric" as the exact value of the distance between two datasets does not matter; only the neighborhoods matter, i.e. the other datasets in the benchmark are ranked depending on their distance w.r.t. the current dataset.

---

### Official Review · Reviewer_cRCu · 2021-11-04

**Correctness:** 3
**Technical Novelty And Significance:** 2
**Empirical Novelty And Significance:** 3
**Recommendation:** 5
**Confidence:** 2

**Main Review:**

The proposed method is based on the exciting idea of extracting the features of the target data as meta-features using optimal transport. For example, strengths and Weaknesses in this paper are respectively given as follows.

Strength.
- AutoML is a hot topic, and the idea of using optimal transport is interesting.
- Experiments on Open ML CC-18 benchmark data show that the proposed method improves the performance of AutoSklearn etc.
The proposed method's speed is acceptable, and the meta-features provide practical hints for using ML algorithms.

Weakness.
The proposed method's effectiveness has been shown empirically, mainly through experiments, but not sufficiently theoretically. Therefore, the limitations of the proposed method are not clear.

**Summary Of The Paper:**

In this paper, the authors address the AutoML problem, which aims to automatically select the best ML algorithm and its hyperparameter configuration for a dataset, and propose an approach to this problem that learns meta-features of the dataset. The proposed method, MetaBu, learns new meta-features by optimal transport according to the space of distributions of hyperparameter configurations. Meta-features in MetaBu is known only once and induce a topology in a set of data sets. Experiments on the OpenML CC-18 benchmark have shown that MetaBu meta-features can improve the performance of the state-of-the-art AutoML systems AutoSklearn and Probabilistic Matrix Factorization. Furthermore, the examination of MetaBu meta-features provides hints on when an ML algorithm will work. Finally, a topology based on MetaBu meta-features can estimate the intrinsic dimension of the OpenML benchmark for a given ML algorithm or pipeline.


**Summary Of The Review:**

This paper proposes a method to extract meta-features based on optimal transport to improve the performance of AutoML, which is a hot topic in the field of machine learning. The usefulness of the method is shown through benchmark data, but the theoretical argumentation is not sufficient.

---

> ### Author Response · Authors · 2021-11-16
> **AutoML as a black-box optimization problem: why empirical validation.**
>
> Q: Weakness. The proposed method's effectiveness has been shown empirically, mainly through experiments, but not sufficiently theoretically.
>
> A: AutoML is an empirical field, as shown by the fact that previous and current AutoML approaches (please see a list of references below) are only assessed empirically.
>
> While AutoML is always handled as a black-box optimization problem, the presented work includes as a novel aspect the interpretability of the approach, through the computation of the intrinsic dimensionality of an AutoML benchmark w.r.t. a given ML algorithm or pipeline $A$, and the highlighting of the different basic features relevant to $A$ performance.
>
> List of references
> ** James Bergstra, R. Bardenet, Yoshua Bengio, and Balázs Kégl. Algorithms for hyper-parameter optimization. NIPS 2011, volume 24
>
> ** Chris Thornton, Frank Hutter, Holger Hoos, and Kevin Leyton-Brown. Auto-WEKA: Combined Selection and Hyperparameter Optimization of Classification Algorithms. KDD 2013, 2013
>
> ** Rémi Bardenet, Mátyás Brendel, Balázs Kégl, and Michèle Sebag. Collaborative hyperparameter tuning. ICML 2013, pp. 199-207.
>
> ** Matthias Feurer, Aaron Klein, Katharina Eggensperger, Jost Springenberg, Manuel Blum, and Frank Hutter. Efficient and robust automated machine learning. NIPS 2015, pp. 2962–2970.
>
> ** Mustafa Misir and Michèle Sebag. Alors: An algorithm recommender system. Artificial Intelligence, 244:291–314, 2017
>
> ** Frank Hutter, Lars Kotthoff, and Joaquin Vanschoren (eds.). Automated Machine Learning: Methods, Systems, Challenges. The Springer Series on Challenges in Machine Learning. Springer, 2019.
>
> ** Isabelle Guyon et al. Analysis of the AutoML challenge series 2015-2018. In Automated Machine Learning: Methods, Systems, Challenges, Springer 2019.
>
> ** Chengrun Yang, Yuji Akimoto, Dae Won Kim, and Madeleine Udell. OBOE: collaborative filtering for AutoML model selection. In ACM SIGKDD International Conference on Knowledge Discovery & Data Mining, pp. 1173–1183. ACM, 2019
>
> ** Herilalaina Rakotoarison, Marc Schoenauer, Michèle Sebag. Automated Machine Learning with Monte-Carlo Tree Search. IJCAI 2019: 3296-3303
>
> ** Matthias Feurer, Aaron Klein, Katharina Eggensperger, Jost Tobias Springenberg, Manuel Blum, Frank Hutter: Auto-sklearn: Efficient and Robust Automated Machine Learning. Automated Machine Learning 2019: 113-134
>
> ** Zhengying Liu et al., Winning Solutions and Post-Challenge Analyses of the ChaLearn AutoDL Challenge 2019. IEEE Trans. Pattern Anal. Mach. Intell. 43(9): 3108-3125 (2021)

---

### Author Response · Authors · 2021-11-16
**General comments**

We thank the reviewers for their thorough reviews. We took into account all comments and suggestions to prepare the revised version.
Additional experiments are reported to assess the impact and significance of the approach, e.g. including pairwise comparison with the baselines with p-value for the significance of the improvement along the iterations.

During the rebuttal period, we added additional plots to emphasize the improvement of our approach; for instance, pairwise comparison with the baselines, p-value curve along the iterations and performance curves. Please find below each review the detailed answers to the direct questions and clarifications as needed.

---

### Decision · Program_Chairs · 2022-01-20

**Decision:**

Accept (Spotlight)

**Comment:**

This paper addresses an important issue of AutoML systems, specifically their ability to "cold start" on a new problem. Some of the reviewers initially had concerns about the experimental validation and the theoretical foundations of the method, but during the discussion phase the authors addressed concerns extremely well. The authors already included most of the feedback of reviewers, further strengthening the paper.